# New structures of Class II Fructose-1,6-Bisphosphatase from *Francisella tularensis* provide a framework for a novel catalytic mechanism for the entire class

Anna I. Selezneva[1], Luke N. M. Harding[2], Hiten J. Gutka[1¤], Farahnaz Movahedzadeh[1,2]*, Celerino Abad-Zapatero[1,3]*

1 Institute for Tuberculosis Research, University of Illinois at Chicago, Chicago, Illinois, United States of America, 2 Department of Pharmaceutical Sciences, University of Illinois at Chicago, Chicago, Illinois, United States of America, 3 Center for Biomolecular Sciences, University of Illinois at Chicago, Chicago, Illinois, United States of America

¤ Current address: Bristol-Myers Squibb, Summit, New Jersey, United States of America
* movahed@uic.edu (FM); caz@uic.edu (CA-Z)

**Data Availability Statement:** All the structural files for the five structures discussed are available from

## Abstract

Class II Fructose-1,6-bisphosphatases (FBPaseII) (EC: 3.1.3.11) are highly conserved essential enzymes in the gluconeogenic pathway of microorganisms. Previous crystallographic studies of FBPasesII provided insights into various inactivated states of the enzyme in different species. Presented here is the first crystal structure of FBPaseII in an active state, solved for the enzyme from *Francisella tularensis* (*Ft*FBPaseII), containing native metal cofactor $Mn^{2+}$ and complexed with catalytic product fructose-6-phosphate (F6P). Another crystal structure of the same enzyme complex is presented in the inactivated state due to the structural changes introduced by crystal packing. Analysis of the interatomic distances among the substrate, product, and divalent metal cations in the catalytic centers of the enzyme led to a revision of the catalytic mechanism suggested previously for class II FBPases. We propose that phosphate-1 is cleaved from the substrate fructose-1,6-bisphosphate (F1,6BP) by T89 in a proximal α-helix backbone (**G88-T89-T90-I91-T92-S93-K94)** in which the substrate transition state is stabilized by the positive dipole of the ⟨-helix backbone. Once cleaved a water molecule found in the active site liberates the inorganic phosphate from T89 completing the catalytic mechanism. Additionally, a crystal structure of *Mycobacterium tuberculosis* FBPaseII (*Mt*FBPaseII) containing a bound F1,6BP is presented to further support the substrate binding and novel catalytic mechanism suggested for this class of enzymes.

## Introduction

Gluconeogenesis is a critical metabolic pathway for the survival of organisms when dietary intake of glucose is insufficient or absent. Class II fructose-1,6-bisphosphatase (FBPaseII, EC: 3.1.3.11) has been shown to be one of the key gluconeogenesis enzymes essential for the

the PDB with accession codes: 7txg, 7txa, 7txb, 8g5w, and 8g5x.

**Funding:** Potts Memorial Foundation (grant No. G3541 to Farahnaz Movahedzadeh, Celerino Abad-Zapatero); Chicago Biomedical Consortium (grant No. 084679-00001 to Farahnaz Movahedzadeh, Celerino Abad-Zapatero). The funders had no role in study design, data collection and analysis, decision to publish, or preparation of the manuscript.

**Competing interests:** The authors have declared that no competing interests exist.

**Abbreviations:** *Ec*FBPase, *E. coli* FBPase; F1,6BP, fructose 1,6 bisphosphate; F6P, fructose-6-phosphate; FBPase, fructose-1,6-bisphophatase; *Ft* $(Mg^{2+})$FBPase, *F. tularensis* FBPase containing a $Mg^{2+}$ divalent cation or crystallized in the presence of such a cation; *Ft*$(Mn^{2+})$FBPase, *F. tularensis* FBPase containing a $Mn^{2+}$ divalent cation or crystallized in the presence of such a cation; *Ft*FBPase, *F. tularensis* FBPase; *Ft*FBPaseII, class II *Ft*FBPase; and similarly for the others; MR, Molecular Replacement; *Mt*FBPase, *Mycobacterium tuberculosis* FBPase; NAD, nicotinamide dinucleotide phosphate; PDB, Protein Data Bank.

virulence of *Francisella tularensis* [1], a highly pathogenic Gram-negative coccobacillus species that causes tularemia in humans worldwide. Due to the potential lethality and rarity of tularemia in mainstream medicine, *F. tularensis* has been recognized as a category A bioterrorism agent [2]. The pathogen replicates intracellularly to high densities mainly in human macrophages [3], utilizing host-derived fatty acids, pyruvate, glycerol and amino acids as a carbon, nitrogen, and energy source for the *de novo* synthesis of molecules that are essential for the pathogen's survival and growth. *F. tularensis* FBPaseII (*Ft*FBPaseII) is encoded by the *glpX* gene, and catalyzes the hydrolysis of F1,6BP to F6P and inorganic phosphate. Thus, survival of the pathogen using non-glucose carbon sources such as glycerol underlies the importance of gluconeogenesis. Furthermore, F6P is a precursor of the pentose phosphate pathway and necessary in the *de novo* synthesis of pentose phosphates, lipopolysaccharides, and peptidoglycan which are all molecules crucial for the pathogen's intracellular survival.

FBPases are a diverse group of enzymes consisting of five structurally different classes (I-V) sharing similarities in the overall organization of the secondary structure elements although having significantly different amino acid sequences [4]. Class I-V FBPases possess two adjacent α/β domains, one of which is two-layered (α-β; CATH code 3.30.540), and the other is three-layered (α-β-α; CATH code 3.40.190). The two domains combine to form a five-layered sandwich-like structure in which the β-sheets are positioned nearly orthogonal to each other. Class-specific differences are determined by the number of strands in each sheet, the boundaries of the strands, and the topology connecting them. In particular, class I consists of two different contiguous β-sheets (A, B) with a total of 13 strands. In contrast, class II FBPases are arranged as an extended 14-stranded β-sheet (AAAA-BBBBBB-AAAA) composed of 4 strands, interrupted in the middle by a distinct β-sheet, (B, 6 strands) and followed by the continuation of the remaining four β-strands of sheet A on the other side. Consequently, the network of amino acid residues congregating to form the active sites of the enzymes result in very different binding of the same substrate F1,6BP in the two classes. The structure of the Class V found in Archaea is different exhibiting a four-layer α–β–α–β sandwich of novel topology and binding the substrate in the open-keto form [5].

Class II FBPases are highly conserved enzymes with over 100 species sharing 50% or higher sequence identity, whereas only about 10% of sequence identity is shared with FBPases of other classes [6]. Prokaryotic FBPases include members of all five classes, while eukaryotic FBPases are mostly limited to class I. This and the fact that no other FBPases have been discovered in *F. tularensis* to this day makes *Ft*FBPaseII an attractive target for the structure-guided drug design of class II selective FBPaseII inhibitors.

Despite the growing number of structures available for class II FBPases, the details of the catalytic mechanism and the distinctions determining metal dependence of the members of the class remain under-investigated. Previous attempts to investigate a catalytic mechanism of FBPaseII involved crystal structures of *Escherichia coli* FBPaseII (*Ec*FBPaseII) [6]; PDB entries 3big, 3bih and 3d1r. The proposed mechanism was based on the *Ec*FBPaseII D61A mutant crystal structure (PDB entry 3d1r). The structure revealed the presence of two $Ca^{2+}$ ions, one $Mg^{2+}$ ion (coordinated in between the two phosphate groups of the substrate on the opposite site) and a molecule of substrate F1,6BP in the active center of the enzyme. However, biochemical characterization of *Ec*FBPaseII by the same authors demonstrated that not only is mutant D61A essentially inactive, but $Mn^{2+}$ (which is absent from the structure) is solely required for *Ec*FBPaseII activity, not $Ca^{2+}$ nor $Mg^{2+}$. The inactivity observed was consistent with the previous observations claiming that replacement of $Mn^{2+}$ with $CaCl_2$ or $MgCl_2$ results in almost complete loss of activity [4,6]. Nevertheless, the authors hypothesized a catalytic mechanism involving two $Ca^{2+}$ ions in the active center: one to stabilize the negative charge on the leaving phosphate group and one to coordinate the

nucleophilic water. The authors acknowledged that the $Mg^{2+}$ ion seems not to be involved in catalysis as it is liganded to the inactive phosphate-6 in the substrate molecule. Hence, the available structure of *Ec*FBPaseII D61A with $Mg^{2+}$ and $Ca^{2+}$ in the active center is not a representation of the active state of the enzyme and does not provide appropriate details towards the mechanism of hydrolysis.

Currently, only one published FBPaseII structure contains catalytic product in a combination with appropriate metal cofactor (PDB entry 6ayu). This is a crystal structure of the partially active T84S mutant of *M. tuberculosis* FBPaseII (*Mt*FBPaseII) complexed with $Mg^{2+}$ and F6P. However, the presence of glycerol in the active center renders it difficult to infer a consistent catalytic mechanism. Our own previous attempts to solve the *Ft*FBPaseII structure resulted in the crystal structure of the $Mg^{2+}$-inactivated enzyme [7]; PDB entry 7js3. This conclusion was based in part on the *Ft*FBPaseII activity studies in the presence of different divalent cations [8]. Similar to *Ec*FBPaseII, it was shown that *Ft*FBPaseII depends solely on $Mn^{2+}$ for activity, while $Mg^{2+}$ and $Ca^{2+}$ do not support catalysis. However, while $Mn^{2+}$ was examined at a broad concentration range, $Mg^{2+}$ and other ions were studied at a single concentration of 100 mM. It was demonstrated [9] that the amplitude of FBPasesII activity may vary significantly depending on the concentration of the metal cofactor in question, ranging from relatively high enzyme activity to its complete loss. Moreover, in some cases, either $Mg^{2+}$ or $Mn^{2+}$ can activate a class II enzyme, while in others, only $Mn^{2+}$ is required for activity. This work presents the investigation of *Ft*FBPaseII activity within a broad range of $Mn^{2+}$ and $Mg^{2+}$ concentrations. There is no evidence that class II FBPases require other metals for catalysis besides $Mg^{2+}$ and $Mn^{2+}$. In contrast, FBPases of other classes can be activated by $Zn^{2+}$, for example [10,11]. Presented in this work is the crystal structure of the wild type *Ft*FBPaseII in its active state containing both F6P and a catalytic metal cofactor $Mn^{2+}$. The structure enabled the identification of the residues responsible for the coordination of both ligands and for modeling the missing phosphate group into the active site. Additionally, for the purposes of discussion, we present the inactive state crystal structures of the wild type *Ft*FBPaseII containing F6P and $Mn^{2+}$ and a serendipitous structure of the wild type *Mt*FBPaseII liganded with substrate F1,6BP.

This work focuses on improving our understanding of the catalytic mechanism of the entire class II FBPases in relation to the better-characterized class I that is predominant in higher organisms, combining analysis of existing and novel structures together with an experimental investigation into the properties of *Mt* and *Ft*FBPasesII. The results provide a framework for a novel catalytic mechanism for this class of enzymes, which will highlight the critical importance of the Thr89-OH (or equivalent) as the attacking nucleophile and the positive charge at the N-terminal dipole of a conserved helix as the stabilizing element of the transition state of the cleavable phosphate.

## Materials and methods

### Gene cloning, production, purification of recombinant *Ft*FBPaseII, and storage

A previously engineered construct of the *F. tularensis glpX* gene containing an N-terminal His6 tag in pET-15b vector was used for overexpression of the *Ft*FBPaseII enzyme in *E. coli* [8]. Overexpression of the wild-type *Ft*FBPasesII and the purification of the proteins from *E. coli* was performed as described previously [7]. The protein was purified, concentrated to at least 10 mg/ml, and stored as single-use aliquots at 193 K until further use in 20 mM Tris–HCl pH 8.0, 50 mM KCl, 10% glycerol. All purification procedures were performed at 277 K.

## Enzymatic assays

Phosphatase activity was quantified spectrophotometrically in Absorbance Units (AU) at 630 nm using colorimetric Malachite Green assay. The reaction mixture (80 μl) contained 40 mM Tris–HCl pH 8.0, 100 mM KCl, 12.5 μg/ml of *Ft*FBPaseII, and 450 μM F1,6BP. MgCl$_2$ and MnCl$_2$ varied as necessary. After 10 min of incubation at 293 K the reaction was quenched by the addition of 20 μl of Malachite Green reagent [12]. Then the color was allowed to develop for an additional 3 min at 293 K. Negative control reactions contained no metal cofactor or enzyme. Positive control contained alternative FBPasesII. All reagents used for protein purification and enzymatic assays were from Sigma.

## Protein crystallization

**Form A**–The complex of *Ft*FBPaseII with Mn$^{2+}$ and F6P was prepared prior to crystallization. The purified protein sample in 20 mM Tris–HCl pH 8.0, 50 mM KCl, and 10% glycerol was sequentially mixed with solution of MnCl$_2$ in 100 mM KCl, and then with solution of F6P in 100 mM KCl. These conditions were chosen to match the ones in which the *Ft*FBPaseII was catalytically active (Fig 2B). The resulting protein concentration in the sample was 11.45 mg/ml, both F6P and MnCl$_2$ were at 1 mM each. Precipitant was added immediately upon the complex preparation. Crystals were grown at 291 K using the hanging drop vapor diffusion method [13], combining the enzyme-containing sample with precipitant in a 2:1-sample:precipitant ratio. The crystal used for diffraction studies was obtained with precipitant containing 8% tacsimate pH 6.0 and 20% polyethylene glycol 3,350 from Hampton Research. The crystal was cryoprotected by a solution of 33% glycerol in the crystallization buffer and flash-frozen in liquid nitrogen.

**Form B**—Prior to crystallization, the sample of purified *Ft*FBPaseII in 20 mM Tris–HCl pH 8.0, 50 mM KCl, and 10% glycerol was mixed with 200 mM MnCl$_2$ solution in 100 mM KCl and then with 4 mM F1,6BP in 100 mM KCl. The conditions for crystallization were chosen to be near those at which the *Ft*FBPaseII was catalytically active (Fig 2B). The resulting protein concentration in the sample was 10 mg/ml, MnCl$_2$ was at 50 mM, and F1,6BP was at 1 mM. Precipitant was added immediately upon the addition of substrate. Crystallization was achieved as described above for Form A and a diffractable crystal was obtained in 0.2 M sodium malonate pH 6.0 and 20% polyethylene glycol 3350 from Hampton Research. The crystal was flash-frozen in liquid nitrogen without cryopreservation.

**Form C**—Crystals of *Mt*FBPaseII complexed with the F1,6BP substrate were obtained by mixing 100 μl of protein solution with 10 μl of 10 mM substrate solution. The protein solution contained 1 mM MgCl$_2$. These conditions were near the ones at which MtFBPaseII was catalytically inactive (Fig 2A). The sample was incubated for 30 min at 303 K and then hanging droplets were formed by the addition of precipitant at 1:1 ratio. Well, diffracting crystals were obtained in 2.9 M sodium malonate pH 4.0 (Malonate grid from Hampton Research). The crystals were typically medium size (20 x 30 x 50 μm) and had a well-defined morphology of hexagonal bipyramids.

**Form D- (Ft-Mn-optimal-Pi).** The purified FtFBPase was prepared as indicate above. The crystallization conditions were as follows: hanging droplet, Hampton Research PEG/Ion2 #14 (8% Tacsimate pH6.0; 20% PEG 3350). Crystals were frozen in a well containing 20% glycerol.

**Form E- (Ft-Mn-suboptimal-F1,6BP).** The approximate suboptimal concentration of the divalent cation to slow down/prevent the catalytic reaction was assessed by performing Malachite

Green activity assays at fine increments of the Mn2+ divalent cation around the minimun concentration required for activity (see above). The successful crystallization conditions were hanging droplet, Hampton Research PEG/Ion2 #30 (0.2M Ammonium Tartrate dibasic pH7.0; 20% PEG 3350). Crystal was frozen in well buffer containing 20% Glycerol.

## Data collection, structure solution and refinement

X-ray diffraction data for both crystals forms A and B were collected at 100 K on the LS-CAT 21-ID-D beamline, using a Geiger 9M detector at the Advanced Photon Source (APS), Argonne National Laboratory, Illinois, USA. All the structures were solved by molecular replacement (MR) using the software MOLREP [14] as implemented in the software suite CCP4 [15]. QtMG [16] was used to produce publication-quality figures.

**Form A**. A wedge of 180˚ of data was collected in fine slicing mode (0.20˚ per image, 900) at a crystal to detector distance of 200 mm with an exposure of 0.07 secs per image. The corresponding structure (Form A) was solved by molecular replacement using the tetramer of the $Mg^{2+}$-containing structure (PDB entry 7js3) as a search model. The data were processed and reduced with HKL2000 [17]. The refinement calculations were done using predominantly PHENIX [18] and alternating with the REFMAC5 [19] implementation in CCP4 [15] to facilitate the interpretation of the very external loops that were often smoothed by the bulk solvent correction in PHENIX. Glycerol molecules were added when putative water molecules did not satisfy the positive (3σ) $F_o$-$F_c$ electron density peaks. Structure rebuilding and revisions were done using COOT as implemented in the PHENIX suite [18]. Indexing of the initial diffraction snapshots suggested that this crystal form was triclinic (P1) with the following cell parameters a = 64.18, b = 76.23, c = 77.92 Å; α = 68.02˚, β = 68.22˚, γ = 76.65˚ diffracting to about 1.9 Å. The size of the unit cell for a P1 lattice suggested that the unit cell of the crystal contained a full tetramer of *Ft*FBPaseII, each chain containing 328 amino acids plus an N-terminal His-tag). Throughout the refinement there were significant differences in the quality and definition of the four different chains near the environment of the metal site waters and active site since non-crystallographic symmetry (NCS) restraints were not used and the solvent exposure and active site accessibility varied widely.

**Form B**. A wedge of 180˚ of data were collected in fine slicing mode (0.25˚ per image, 720) at a crystal to detector distance of 250 mm with an exposure of 0.08 secs per image. The data were processed and reduced with HKL2000 [17]. The structure (Form B) was solved by molecular replacement using a partially refined model of Form A, confirming the presence of two *Ft*FBPaseII tetramers in very similar orientations separated by about 70 Å along the crystal c axis. The refinement protocols were analogous to the ones used for Form A. Indexing of the initial diffraction snapshots suggested that the crystal form was also P1 with cell parameters a = 65.82, b = 76.37, c = 141.11 Å; α = 76.98˚, β = 87.10˚, γ = 75.84˚ and diffracting only to about 2.4 Å. These crystals were more radiation sensitive than those of form A and thus a compromise between data quality and exposure time was adopted. The size and symmetry of the unit cell, when compared to form A, suggested that the crystal cell contained two tetramers of *Ft*FBPaseII (328 amino acids plus an N-terminal His-tag). The conformation of most external loops varied quite significantly among different chains, possibly explaining the low symmetry of both crystal forms.

**Form C**. These crystals were obtained at the earlier stages of the *Mt*FBPaseII project and since they diffracted only to about 3.7 Å resolution, they were not fully analyzed at the time. A wedge of 200˚ of data was collected using 1.0˚ per frame strategy. The data was collected on

a MarCCD 300 detector at the SER-CAT beamline at the APS. The data was processed and reduced with HKL2000 [17]. The structure (Form C) was initially solved by molecular replacement using one chain of the *Mt*FBPaseII structure (PDB entry 6ayu), yielding two separate chains in the asymmetric unit. However, refinement was not pursued at that time. An initial analysis of these crystals showed that they were the same hexagonal space group P6₁22 as the previously analyzed *Mt*FBPaseII crystals [20] (PDB entries 6ayy, 6ayu, 6ayv) but with an approximately double c axis (a = b = 117.1, c = 325.92 Å, vs. c = 140 Å) diffracting only to about 3.7Å. The interest in these crystals was renewed when a preliminary Molecular Replacement (MR) solution of the structure suggested the presence of the substrate F1,6BP in a very loose packing arrangement of *Mt*FBPaseII tetramers. The initial MR solution was later confirmed, and the structure was fully refined. Given the limited resolution of the data, the refinement proceeded initially with NCS restrains as the main aim of the analysis was to provide a robust electron density to fit the substrate in the active site pocket of the two chains in the asymmetric unit. Later the NCS restraints were removed, and refinement proceeded with conventional protocols.

**Form D (Ft-Mn-optimal-Pi).** Initial indexing at the beam line showed that the crystal was triclinic (P1) thus a rapid coverage of the full reciprocal lattice was attempted. A wedge of 360˚ of data was collected with 1˚ per frame (360 frames) with a 3 second exposure in a MarCCD 300 detector with a crystal to detector distance of 275 mm at the LS-CAT beamline at the APS. The final integration cell parameters were a = 63.84, b = 75.88, c = 77.62 Å; α = 67.83˚, β = 68.14˚, γ = 76.37˚ with the crystal diffracting to about 2.0 Å. The structure was solved by molecular replacement using the available PDB entry 7js3 of the same enzyme containing a tetramer in the asymmetric unit.

**Form E (Ft-Mn-suboptimal-F1,6BP).** Forms A, D and E are essentially isomorphous with minor differences in the unit cell parameters. A wedge of 200˚ of data was collected with 0.5˚ per frame (400 frames) and three seconds exposure at a crystal to detector distance of 300 mm. Unit cell integration parameters were a = 63.76, b = 74.58, c = 77.19 Å; α = 67.89˚, β = 67.89˚, γ = 77.01˚ with significant diffraction to about 2.2 Å. The structured was also solved independently by molecular replacement using the full tetramer in PDB entry 7js3. Initial stages of refinement were done by standard protocols, with the quality of the structure and density of Chain C being better than the other three chains. The electron density of the maps ($2F_o$-$F_c$/$F_o$-$F_c$) maps at the active site of this chain suggested the presence of a partially occupied substrate molecule (Fig 1A). Thus, a polder map [21] was calculated at two different resolutions to conclusively confirm if the substrate (Fructose-1,6-Bisphosphate) was present. The cross-correlation values for the three calculated maps (2.5 Å) were: CC(1,2) = 0.6487, CC(1,3) = 0.8791, CC(2,3) = 0.593; strongly suggesting the presence of the ligand. The resulting polder map is shown in Fig 1B. Details of the data collection, reduction, and structure refinement for all five crystal forms are summarized in Table 1.

## Results and discussion

### *Ft*FBPaseII and *Mt*FBPaseII activity at various $Mn^{2+}$ and $Mg^{2+}$ concentrations

Initial characterization of the divalent metal cation requirements for the *Mt*FBPaseII enzyme suggested that at low (<3 mM) $Mn^{2+}$ concentrations the enzyme could exhibit a 'burst of activity' while high concentrations would significantly inhibit the activity (Fig 2A). The binding of the genuine cofactor $Mg^{2+}$ exhibit the hyperbolic profile of activity vs. concentration

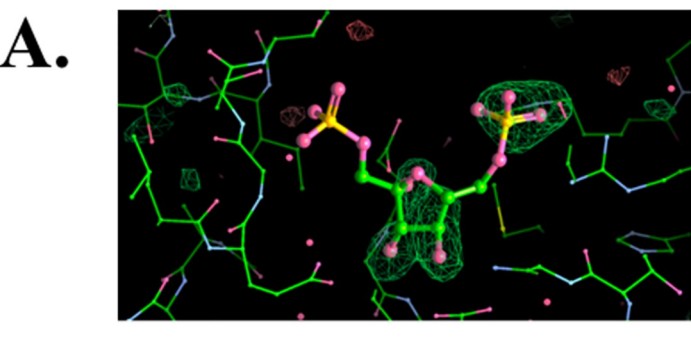

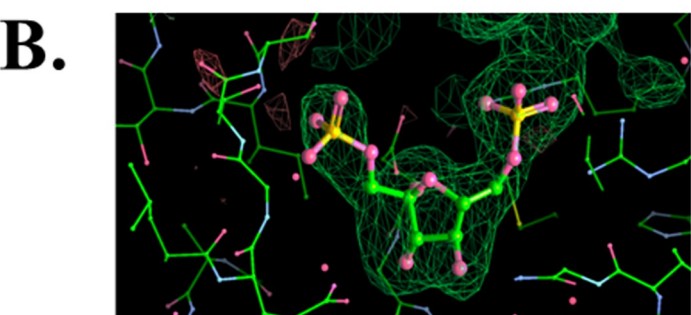

**Fig 1. Electron density maps of active site of Chain C in the crystal structure of Form E.** (A) Omit map. (B) Corresponding polder map contoured 3σ level. The density of the cleavable phosphate (Phosphate-1, left) is appreciably weaker even in the polder map suggesting it might be partially occupied (or cleaved) in some tetramers within the triclinic crystal.

upon saturation of the metal binding site(s). In contrast, for *Ft*FBPaseII, the data show (Fig 2B) that the activity of *Ft*FBPaseII stays high (1.25 ± 0.25 AU) at $Mn^{2+}$ concentrations ranging from 1 mM to 50 mM, whereas the same concentrations of $Mg^{2+}$ cause a complete loss of activity. These results complement previous studies of *Ft*FBPaseII activity conducted in a buffer lacking KCl, where a gradual increase in activity of the enzyme was observed over a similar range of concentrations of $Mn^{2+}$ with a maximum at 50 mM [7]. $Mg^{2+}$ was found to be inhibitory at any concentration tried regardless of KCl absence or presence, supporting the notion that the structure of *Ft*FBPaseII complexed with $Mg^{2+}$ that had been reported earlier is of the inactive enzyme [7].

A similar 'burst of activity' in the presence of low (~2 mM) concentrations of $Mn^{2+}$ has been observed in the activity of the dual Fructose-1,6/sedoheptulose-1,7-bisphophatase (FBP/SBpase) from the cyanobacteria *Synechocystis* (PCC6803) [9]. Synchronous fluorescence and fluorescence quenching studies on the binding of metal ions were used in combination with experimental binding studies to compare the affinities and fingerprints of various metal cations ($Mg^{2+},$ $Mn^{2+},$ $Zn^{2+},$ $Ca^{2+}$) upon enzyme binding. The results independently confirmed the preference for $Mg^{2+}$ and $Mn^{2+}$ over the other two and showed a similar profile to the one shown in Fig 2A, upon variation of the divalent metal concentration. Estimation of the free energy of binding (ΔG) from the slope of the van't Hoff plots showed a significant difference between the binding of $Mg^{2+}$ (-22.8 kJ mol$^{-1}$) vs $Mn^{2+}$ (-15.5) at T = 295 K. More significantly, the binding of $Mn^{2+}$ had a rather large and positive entropy value (205.5 J mol $K^{-1}$) when compared with $Mg^{2+}$ (-467.6 J mol $K^{-1}$) [9]. These quantitative results strongly support the notion that the native divalent cation for PCC6803 is $Mg^{2+}$. In view of the results presented for

**Table 1. Data reduction and refinement statistics for the five crystal structures.**

| | *Ft*(Mn²⁺)FBPase-F6P (Form A, 7txg) | *Ft*(Mn²⁺)FBPase-F6P (Form B, 7txa) | *Mt*FBPase-FBP (Form C, 7txb) | Ft-Mn-Pi (Form D, 8g5w) | Ft-Mn-FBP (Form E, 8g5x) |
|---|---|---|---|---|---|
| Wavelength (Å) | 1.033 | 0.9787 | 1.0000 | 0.97872 | 0.97872 |
| Resolution ranges | 19.92–1.90 (1.97–1.90) | 39.1–2.40 (2.48–2.40) | 20.00–3.71 (3.84–3.71) | 35.559–2.0 (2.07–2.0) | 37.18–2.2 (2.28–2.2) |
| Space group | P 1 | P 1 | P 6₁ 2 2 | P1 | P1 |
| Unit cell (Å, °) (a,b,c) (α, β, γ) | 64.18 76.23 77.9 68.02 68.22 76.65 | 66.39 76.63 141.46 76.839 87.235 75.707 | 117.11 117.11 325.92 90 90 120 | 63.84 75.88 77.62 67.83 68.14 76.37 | 63.76 74.58 77.19 67.83 68.14 76.37 |
| Total reflections | 157404 (3562) | 302628 (4997) | 270665 (1285) | 327058 | 130001 |
| Unique reflections | 85208 (8144) | 101338 (9973) | 14655 (656) | 82358 (8095) | 59757 (5771) |
| Multiplicity | 1.80 (1.8) | 3.0 (3.0) | 18.4 (10.8) | 3.9 (3.7) | 2.2 (2.1) |
| Completeness (%) | 85.87 (82.50) | 98.38 (97.03) | 98.64 (89.76) | 97.98 (96.89) | 96.88 (93.69) |
| Mean I/sigma(I) | 25.27 (4.29) | 17.55 (1.78) | 11.0 (1.60) | 24.86 (2.91) | 20.92 (2.4) |
| Wilson B-factor (Å²) | 27.3 | 42.2 | 86.9[b] | 28.15 | 37.3 |
| R-merge* | 0.054 (0.156) | 0.043 (0.43) | 0.207 (0.678) | 0.055 (0.506) | 0.046 (0.310) |
| R-meas | 0.077 (0.219) | 0.075 (0.604) | 0.140 (0.427) | 0.086 (0.613) | 0.080 (5.505) |
| R-pim | 0.054 (0.155) | 0.042 (0.348) | n/a** | 0.043 (0.315) | 0.054 (0.341) |
| CC1/2 | 0.996 (0.939) | 0.995 (0.692) | n/a** | 0.993 (0.844 | 0.997 (0.779) |
| Reflections used in refinement | 85190 (8141) | 101312 (9973) | 14520 (1280) | 82338 (8093) | 59739 (5771) |
| Reflections used for R-free | 4209 (390) | 5065 (527) | 744 (66) | 4080 (416) | 2943 (290) |
| R-work | 0.187 (0.214) | 0.196 (0.248) | 0.217 (0.289) | 0.1607 (0.1913) | 0.1648 (0.2118) |
| R-free | 0.221 (0.271) | 0.248 (0.328) | 0.273 (0.365) | 0.2046 (0.2532) | 0.2267 (0.2842 |
| Number of non-hydrogen atoms | 10493 | 19787 | 4587 | 10353 | 10337 |
| Macromolecules | 9705 | 19787 | 4513 | 9682 | 9678 |
| Ligands | 51 | 43 | 67 | 112 | 102 |
| Solvent | 573 | 397 | 139 | 559 | 557 |
| Protein residues | 1309 | 2612 | 759 | 1305 | 13040 |
| RMS (bonds) | 0.010 | 0.009 | 0.004 | 0.015 | 0.017 |
| RMS (angles, °) | 1.53 | 1.21 | 0.82 | 1.43 | 1.71 |
| Ramachandran favored (%) | 97.62 | 95.30 | 88.01 | 98.46 | 98.07 |
| Ramachandran allowed (%) | 2.15 | 4.01 | 10.9 | 1.54 | 1.85 |
| Ramachandran outliers (%) | 0.23 | 0.69 | 0.99 | 0.0 | 0.08 |
| Rotamer outliers (%) | 0.00 | 0.00 | 0.00 | 0.0 | 0.10 |
| Clash score | 8.98 | 9.1 | 9.6 | 8.42 | 11.90 |
| Average B-factor (Å²) | 42.21 | 51.87 | 88.03 | 40.77 | 48.13 |
| Macromolecules (Å²) | 42.29 | 52.00 | 88.35 | 40.53 | 47.64 |
| Ligands (Å²) | 54.28 | 70.95 | 89.65 | 60.32 | 71.1 |
| Solvent (Å²) | 39.69 | 43.48 | 77.0 | 41.1 | 52.37 |
| Molprobity score[a] | 1.56[a] | 1.81 | 2.1[a] | 1.39 | 1.56 |

*Statistics for the highest-resolution shell are shown in parentheses. Data reduction parameters have standard definitions.

**n/a. Not available. The data were collected and processed from the crystal forms obtained in the early stages of the project when the data processing software did not provide those data reduction statistics. For structure deposition, the PDB does not require those statistics. $R_{mer}$ and $R_{meas}$ are sufficient.

[a]In spite of the overall limited completion of the data for 7txg (~85%) and lower resolution for 7txb (3.70 Å), the overall Molprobity scores for these two structures (1.56, 2.1) is significantly lower than the nominal resolution, 1.8 and 3.7 Å respectively, supporting the quality of the final refined structures.

[b]The high value of the Wilson B-factor is consistent with the high solvent content (71.7%) and low-resolution diffraction of this crystal form. However, it is included in this work to confirm the location and orientation of the uncleaved substrate (F1,6BP) in the active site of MtFBPase. It is rare to find a substrate bound in the active site of a native enzyme. The resolution and quality of this structure are sufficient to document this finding.

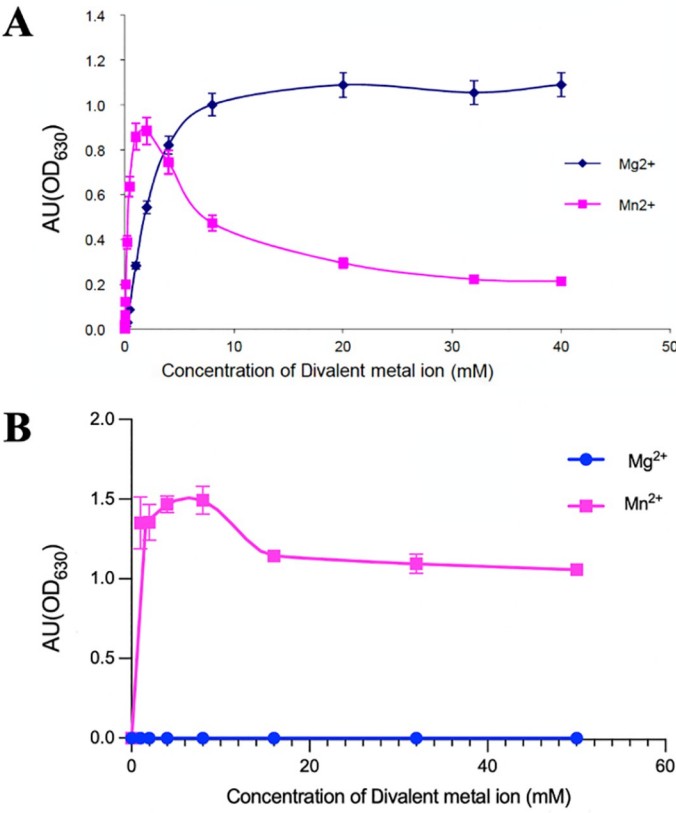

**Fig 2. *Mt*FBPaseII and *Ft*BPaseII activities at difference concentration of divalent cations Mn$^{2+}$ and Mg$^{2+}$.**
Activity profiles of *Mt*FBPaseII (A) and *Ft*FBPaseII (B) at different divalent cation concentrations. Assay conditions as specified in Materials and Methods.

*Mt*FBPaseII here and the previous structures (6ayy, 6ayu, 6ayv), we hypothesize that the binding of Mn$^{2+}$ for *Mt*FBPaseII at low concentration is transient and does not represent the native cofactor binding in the active site. Conversely, the binding of the non-native Mg$^{2+}$ observed in *Ft*FBPaseII [7] represented a fully inactive conformation of the enzyme.

## Overall view of the crystal structures

The four crystal forms (A, B, D, and E) of the *Ft*FBPaseII enzyme were triclinic, containing full tetramers as the macromolecular assembly in the crystals. Forms A, D, and E were isomorphous with one tetramer in the unit cell, while in Form B the b cell axis was doubled and contained two tetramers in the cell with slightly different orientations and displaced by about 40 Ångstroms along the cell axis. Although the polypeptide fold in all the different structures is conserved, this implies a wide variety in the environment of the four (eight in form B) different active sites per tetramer within the crystal and across the crystals, given the different crystallization conditions. The most important features of each crystal structure are reviewed in relation to providing insights into the native metal cofactors of the class, the most consistent active site structure (including coordination waters), and the implications for the catalytic mechanism of the class II FBPases.

**_Ft_(Mn$^{2+}$)FBPaseII-Glycerol/F6P complex (Form A).** The high resolution (1.9 Å) of this structure revealed one approximate 222 (D$_2$) tetramer in the triclinic cell with subtle

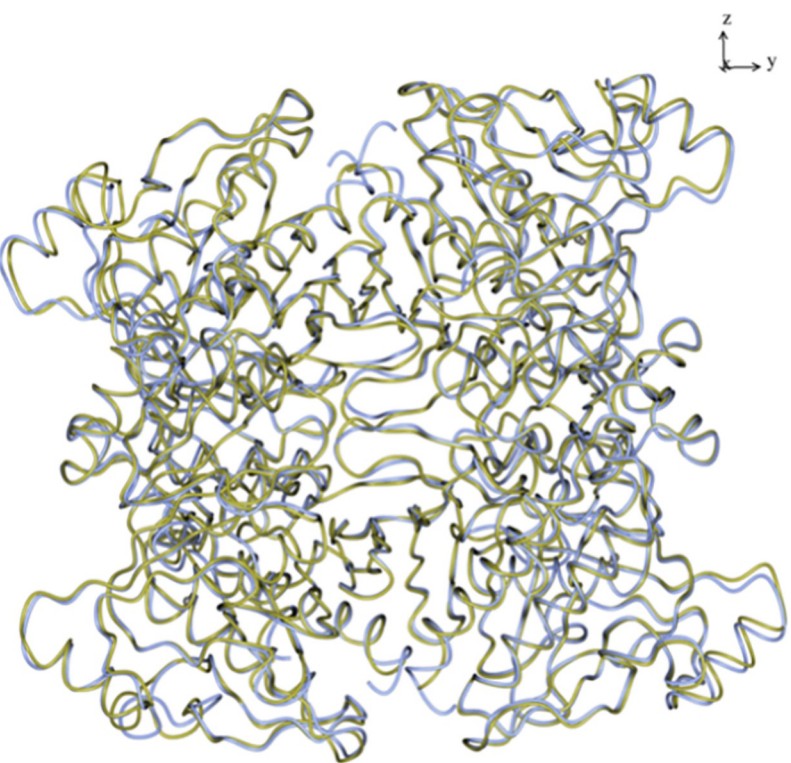

**Fig 3. Superposition of the D$_2$ tetramer of the *Ft*(Mg$^{2+}$)FBPaseII (PDB 7js3) (gold) and the corresponding oligomer of *Ft*(Mn$^{2+}$)FBPaseII-F6P, Form A (ice blue).** The relative orientation of the four chains in the tetramer is conserved, and only relatively minor local displacements are observed in the outermost loops of the tetramer. The inhibition of the *Ft*FBPaseII by the Mg$^{2+}$ cation does not affect the overall structure of the tetramer.

differences among the four chains. The overall quaternary arrangement of the four chains did not differ significantly from the tetramer described earlier for the Mg$^{2+}$ liganded structure (PDB entry 7js3). However, significant local differences were observed in chains B and D, particularly in the residues related to the orientation and disposition of the protruding helix (H12, Chain C Phe235-Leu252; Chain D Ala231-Leu261) (Fig 3). Both were involved in crystal contacts on this triclinic form. The tertiary structures did not differ significantly from the previously reported structure however, chains A and B showed weak density for the residues in the protruding helix Ala234-Ala248 and chain D in the external residues Gly56-Met65 of the Ψ loop [8]. All the mentioned regions also showed significant local deviations from the NCS using subunit A as a reference.

Several discrete glycerol molecules were refined to reasonable B factors (<80 Å$^2$) at distinct locations in the four chains A(1), B(5), C(2), and D(4). The phosphate groups were refined at the putative locations of the product F6P (residues Arg165-Pro-Arg167) in chains B, C, and D. Attempts to refine full F6P molecules at these locations failed because of the presence of well refined glycerol molecules that competed with the full F6P occupancy (Fig 4). The primary sequence motif in FtFBPaseII 184IxDGD187 conserved in all members of this class and involved in the binding of the furanose ring hydroxyls (see below) is critical to bind glycerol at this site.

A chemically consistent environment for the Mn$^{2+}$ cofactor in this crystal structure was refined in the proximity of the F6P for chain C (Figs 5 and 6A). The structure revealed an octahedrally coordinated Mn$^{2+}$ ion, surrounded by the side chain carboxylate group of Asp84, the

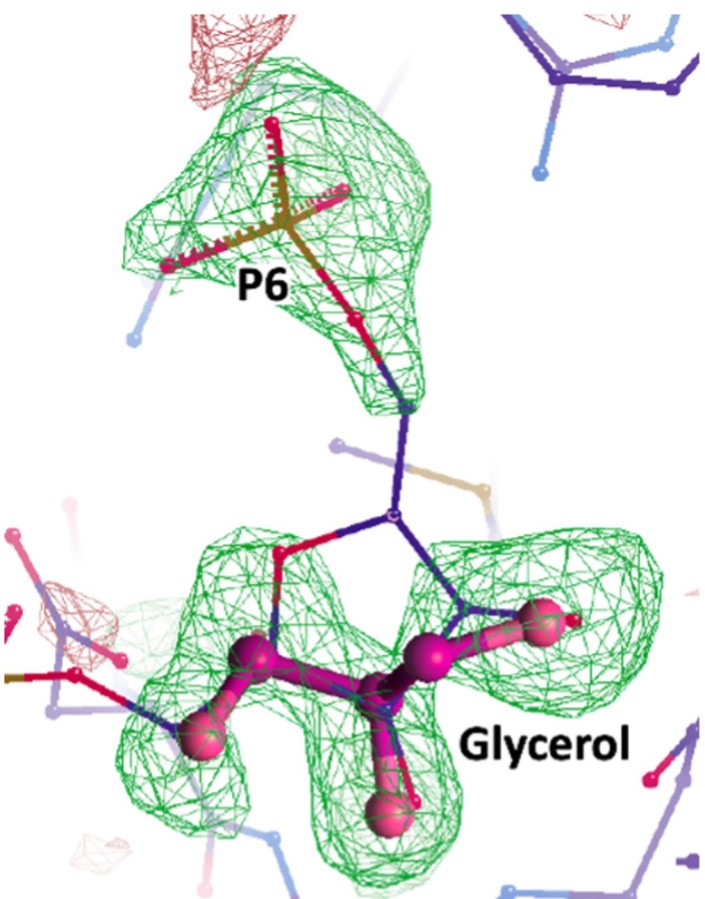

**Fig 4. Omit electron density map of the refined structure of *Ft*FBPaseII (Chain C).** Omit map centered on the product binding region showing the partial occupancy of F6P and the corresponding fitting of a bound glycerol molecule.

main chain carbonyl oxygen of Leu86, and four water molecules. In addition, the side chain of Glu57 projects its carboxylate group towards the lower apical liganded water (distance 3.0 Å) and could play a role in the coordination of the metal since this portion of the chain is disordered in crystals of *Ft*FBPaseII grown in the absence of metal cations ($Mg^{2+}$ or $Mn^{2+}$, unpublished observations). This is also consistent with the various forms of chain disorder in the region of the polypeptide chain between 57EGELDEAP64 in the different crystal structures reported here. The low concentration of divalent cation $Mn^{2+}$ does not permit the interaction of the E57 ligand and the region is partially or fully disordered, depending on the crystal environment of the different subunits.

*Ft*($Mn^{2+}$)FBPaseII-F6P complex (Form B)—Form B represents the most complex crystallographic structure of *Ft*FBPaseII observed to date. The triclinic cell contained two approximate 222 ($D_2$) tetramers separated by about 70 Å, with the orientation of their respective local dyads being slightly different. The overall structures of the two tetramers did not differ substantially as compared to the tetramer observed in the $Mg^{2+}$ containing structure (PDB 7js3) although—as discussed before for the Form A crystals—there were local differences in the external loops involved in the crystal contacts. The most striking observation of this crystal form was the presence of two $Mn^{2+}$ sites in chain D of the first tetramer in this

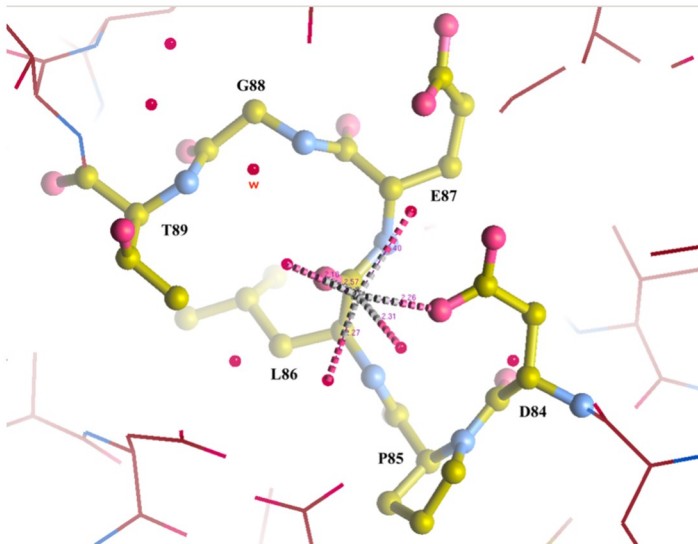

**Fig 5. Approximate distances and residues involved in the coordination of Mn$^{2+}$ in the best-defined (chain C) active site of Form A (7txg).** The image illustrates the canonical, high affinity, M1 site for the native cation of *Ft*FBPaseII. The distances of the water molecules coordinating the Mn$^{2+}$ cation range from 2.2–2.6 Å). The small red sphere labeled W in red is one of the waters displaced by the substrate binding (see below).

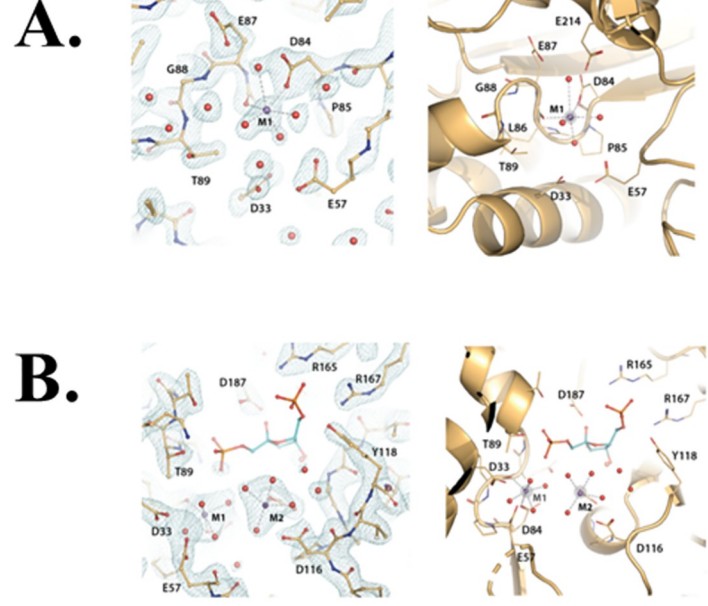

**Fig 6. Divalent metal binding sites in class II FBPases.** (A) Consistent binding mode of Mn$^{2+}$ at site one (M1) in several of the active sites of the structures reported here (e.g. 7txg, chain C, 1.9 Å resolution) and also observed in the previously published structure of *Ft*(Mg$^{2+}$)FBPase (PDB: 7js3). (B) Coexistence of two Mn$^{2+}$ metal sites in the active site of *Ft*FBPase. Refined structure of the active site FtFBPase in Chain D of form B (7txa, chain D) with the two metal sites occupied by the activating cation Mn$^{2+}$. The substrate molecule from form E (8g5x) chain C has been superposed based on the corresponding protein structure. Left 2F$_o$-F$_c$ electron density maps contoured at 1.5σ level. Right the atomic rendering of the structure derived from the corresponding maps. This could represent the active conformation of the FtFBPaseII enzyme posed for a two-metal catalytic mechanism at high Mn$^{2+}$ concentrations.

crystal form (Form B). This is illustrated in Fig 6 where the two-metal binding 'motifs' are contrasted. The best defined and previously observed divalent cation binding motif comprising the contiguous residues 84DPLEGTT90 (Fig 6A) in contrast to the more rare, weaker site, also observed in the $Ft$(Mg$^{2+}$)FBPase (PDB 7js3). This second site only comprises residues Asp116 and E214 with the divalent cation (Mg$^{2+}$/Mn$^{2+}$) in between them. In the previously reported structure containing Mg$^{2+}$, the two carboxylate side chains are closer together, probably due to the smaller size of the Mg$^{2+}$ cation. Since the E214 residue is essentially at the same position (within experimental error), it is Asp116 that extends out significantly (distance E214-D116 OD1 2.8 Å). Consequently, the interaction of residue Tyr118 with the PO$_4$(6) of the substrate is altered.

*Mt*(Mg$^{2+}$)FBPaseII-F1,6BP complex (Form C: 7txb))—The MR solution of this crystal form placed two separate chains (A, B) in the crystal asymmetric unit. The final refined structure confirmed the presence of the F1,6BP substrate in both chains and was used as a reference in conjunction with the *E. coli* structure (PDB 3d1r) to confirm the position and orientation of the substrate in the active site. The position of the substrate phosphate groups between the two structures differed only by approximately 0.7 Å (Figs 1 and S2) and was considered in reasonable agreement to support the proposed enzymatic mechanism for *Mt*FBPaseII, *Ec*FBPaseII, *Ft*FBPaseII and homologous class II FBPases, given the lower resolution of the first structure (see Table 1). The two separate chains appear to have a Mg$^{2+}$ cation bound partially by the phosphate-6 of the substrate, but the resolution of the structure does not permit any further inferences. In the final stages of refinement, it was possible to reassemble the asymmetric unit of the crystal to form a dimer of the two separate chains, homologous to the one suggested in the initial *E. coli* structure but different from the one found in the asymmetric unit of the PDB entry 6ayu. The two different combined dimers form the consensus tetrameric structure of the class II FBPases are formed by crystallographic symmetry, as discussed previously [20]. All four sites were occupied by the substrate F1,6BP. We hypothesized that the observation of the uncleaved substrate in this crystal form was due to the limited availability of the native divalent cation (Mg$^{2+}$) since the concentration was only 1 mM in the crystallization media and the crystal structures do not provide enough evidence for full occupancy of the sites (Fig 2A). Possibly, the pH of the crystallization media (pH = 4.0) could have had an effect since the enzyme at low pH is much less active.

*Ft*(Mn$^{2+}$)FBPase-Pi (Form D—8g5w). The crystallization conditions of this form included high inorganic phosphate concentration in an attempt to capture phosphate ions as reaction intermediates in the active site. One prominent water molecule in the active site of Chain D with high electron density peaks during the refinement progress was initially modelled as phosphate. However, the refinement did not confirm its presence. Nevertheless, in the environment of this active site, a consistent group of three water molecules was found mimicking the presence of the three hydroxyl groups of the furanose ring of the substrate and—significantly—the oxygens around the cleavable phosphate group near Thr89 at the N-terminal of a conserved helix in the active site (Fig 7).

This insight, combined with the observation of the substrate FBP in form E assisted in establishing the consistent binding of the substrate in the active site. In addition, the active site in this chain also reproduced the 'canonical' binding (M1) site of the metal cofactor (Mn$^{2+}$) and the corresponding coordinated waters observed before. Significantly, only one metal cofactor site was occupied in the site that we consider to be the 'high affinity' site defined by

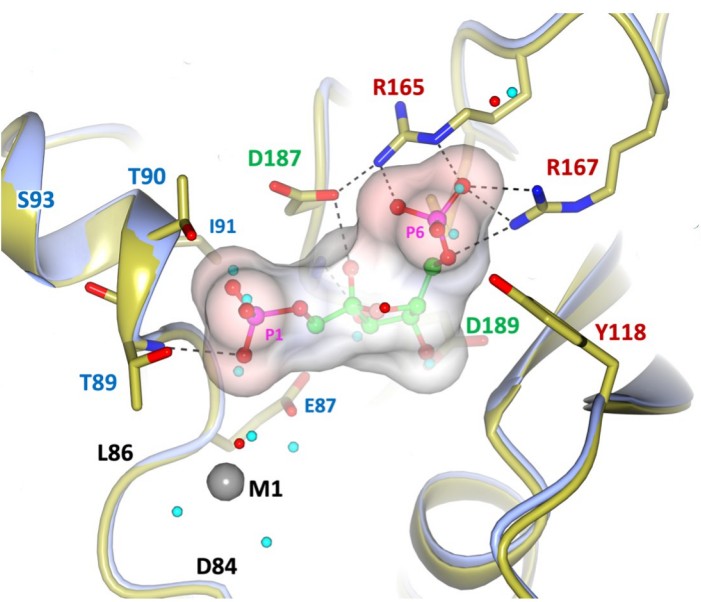

**Fig 7. Active site of Form D *Ft*FBPase crystals.** Structure of Form D (chain C) showing the distribution of water molecules (light blue spheres) within the active site and metal coordination. The position and orientation of F1,6BP derived from the 'polder' map of the form E (chain C) is shown as balls and sticks. Upon binding, the 1-P of the substrate displaces the corresponding water molecules and is within interacting distance with the OH group of Thr89 (dashed line). See S3 Fig for the corresponding electron density for the location of the experimental water molecules. The color coding of the amino acid sequence numbers corresponds to the catalytic elements identified below (**Fig 9**).

the contiguous residues 83LDPLE87 in *Ft*FBPase and 78VDPID82 in other class II FBPases (e.g. MtFBPase) (Fig 5).

*Ft*(low Mn$^{2+}$)FBPase-FBP (Form E—8g5x). The crystallization conditions of this form were selected with the objective of capturing the binding of the substrate to the active site in its native conformation prior to the hydrolysis step. Only in the last stages of the refinement a promising electron density in the active site of chain C could be interpreted as what appeared to be a substrate molecule bound. The omit maps were consistent with the interpretation, but it was only when the $F_o$-$F_c$ maps corrected by bulk solvent were calculated (polder maps) that the evidence was unequivocal (Fig 1). This significant result provided a sound basis for the ensuing characterization of the active site with the substrate bound and the metal cofactor environment near it without the need of any modelling.

### Elements of the catalytic mechanism

The new structures of class II *Ft*FBPase combined with the pre-existing structures of *Mt*FBPases and *Ec*FBPaseII provide a sound basis for characterization of the key elements of the phosphoryl transfer reaction taking place in class II FBPases: i) Metal cofactor binding; ii) substrate recognition and binding, iii) identification, positioning, and activation of the nucleophile; and iv) stabilization of the leaving group [22]. These elements have been illustrated in Fig 9 as inferred from the crystal structures described earlier for the FtFBPaseII. A separate section follows for each of the components.

**Metal cofactor binding.** The detailed analysis of the enzymatic activity of *Ft*FBPaseII and *Mt*FBPaseII in the presence of their apparent metal cofactors (Mg$^{2+}$ andMn$^{2+}$), in

combination with the available structural data and the five additional structures reported here, clarifies the issues of metal specificity vs. metal inhibition for this class of enzymes. Namely, only one high affinity metal (M1) site, basically determined by the continuous residues DPIDGT is required for the hydrolysis of the cleavable phosphate ($PO_4$-1) from the substrate. The presence of the divalent cation at M1 orders the polypeptide chain in the region of residues GEGELDEAPM, with residue E57 binding the divalent cation as the lower apical octahedral ligand. Interestingly, the structural data presented here characterizes the existence of a second (weaker) metal site (M2) that comprises essentially only two non-contiguous residues D116 and E214. We hypothesize that the interplay between these two metal sites results in the unique response of each individual member of the class II FBPases to different concentrations of the two reported cofactors $Mg^{2+}$ and $Mn^{2+}$ in the available publications. The suggestion is made that although in some cases the distinct preference for either one of them is clear (*E. coli*, *F. tularensis* $Mn^{2+}$ vs $Mg^{2+}$), other members could be enzymatically active with either one of them, although at different catalytic rates. Based on the observations presented in this work (Figs 2 and 6B), it could be also possible that even with the preferred metal cofactor (e.g. $Mn^{2+}$, *F. tularensis*) the *Ft*FBPaseII could catalyze the reaction at two different rates depending on whether the active site contains one or two $Mn^{2+}$ sites, at higher cation concentrations.

**Substrate recognition and binding.**   In the class II FBPases previously analyzed structurally (*Ec*FBPaseII: E3d1r; *Mt*FBPaseII: 6ayu, 6ayv; *Ft*FBPaseII: 7js3), the binding of the $PO_4$-6 phosphate is characterized by an Arg-Pro-Arg motif (except *Ec*FBPaseII, Lys-Pro-Arg) with the positively charged side chain groups holding the anchoring phosphate group. Significantly, approximately twenty amino acids down the polypetide chain of all the class II FBPases, there is a strictly conserved motif (IxDGD) that appears to be critical for binding the three hydroxyl groups in the furanose ring of F1,6BP. This binding mode of the substrate is indeed different from the more extensively characterized class I ($Mg^{2+}$ dependent) FBPases where the binding of the anchoring phosphate group ($PO_4$-6) is dominated by Tyr residues (Tyr215, Tyr244, Tyr264) and Asn212. The residues interacting with the furanose ring hydroxyls are Met248 and Asp121. The residues and chemical environment around the cleavable phosphate (P1) have been characterized in more detail in the allosteric ($Mg^{2+}$ dependent) class I FBPase in the two conformational states R and T, through the work on the porcine kidney and rabbit liver enzymes [23,24] (PDB entries: (PDB entries: 1cnq, 1eyi, 1eyj, 1eyk). The structural results presented here strongly suggest that the unique binding mode of the close furanose ring of the F1,6BP substrate is a distinct characteristic of the Class II FBPases and results in a unique reaction mechanism, related to their unique divalent cation specificities.

**Identification, positioning, and activation of the nucleophile.**   Our observations are fully consistent with the previously demonstrated critical role of the hydroxyl side-chain of Thr89 in the catalytic mechanism, where the T89S variant of *Ft*FBPaseII is active (albeit slower). However, the T89A is completely inactive [8]. The T89/S89, amino acid residue, is critical for catalysis but does not provide any significant binding affinity for the substrate. Mutation of the homologous residues of the *Mt*FBPaseII enzyme exhibit similar enzymatic properties (T84S is partially active, but T84A is completely inactive) [20]. A possible structural explanation of this effect can be surmised based on the additional steric hindrance of the side-chain of the Thr84 residue provided by the neighboring residue Leu86 (Ile87 in *E. coli*), facilitating the correct positioning and orientation of the critical hydroxyl group (Fig 8).

**Stabilization of the leaving group.**   The N-terminus of a two turn α-helix (H3; G88-T89-T90-I91-T92-S93-K94) is facing $PO_4$-1, about 3.1 Å away and identifies a structural

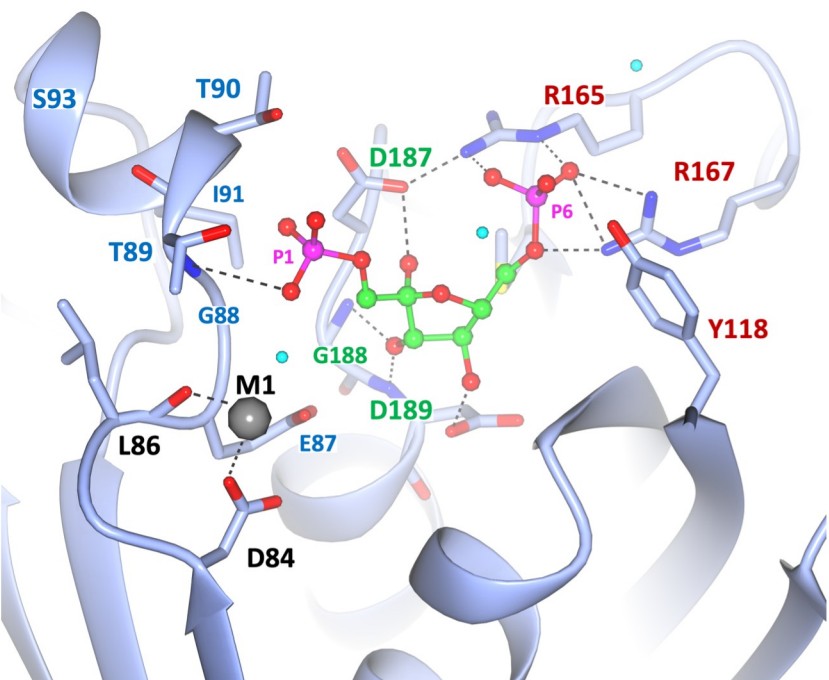

**Fig 8. Interactions of the substrate (F1,6BP) in the active site of *Ft*FBPase (chain C).** The active site is derived from the fitting and refinement of the 'polder' electron density map. Hydrogen bonds are highlighted with dashed lines. Note the position of the Mn²⁺ cation (M1) at the high affinity site and the interactions with the the N-terminal amide protons of the helix (only the closest one is noted for clarity). Hydrogen bond distances noted are within 2.4–3.2 Å.

feature that could provide an excellent 'anchor' to stabilize the phosphate leaving group (P1) (Fig 8). The phosphate binding helix assists in the binding of F1,6BP in the active site and stabilizes the negatively charged tetrahedral transition states of PO₄-1 after the nucleophilic addition Thr89. The helix further decreases the rate of the reverse reaction by coordinating the inorganic phosphate upon hydrolysis, (Figs 9–11). This helix dipole moment arises from the approximate 3.5 Debyes dipole moment generated when the formal charges of the oxygens and nitrogens in each peptide bond neatly align along the helix axis. The strength of the field is proportional to the length of the helix but tapers after two turns which is the size of the helix found in the active site [25]. This is not a unique phenomenon found in class II FBPases as the enzymes lactate dehydrogenase, glyceraldehyde phosphate dehydrogenase, and triosephosphate isomerase all contain phosphate binding helices that at the N-terminus bind to the negatively charged phosphate groups of NAD and/or glyceraldehyde-3-phosphate. The dipole moment has the advantage of having the effect of an isolated charge without being excessively solvated or in the proximity of a salt bridge, as is typical for charged residues [25]. The helical structure stressed above is conserved in all class II FBPases for which the three-dimensional structure is known. The amino acid sequences, if not fully conserved, do not have any insertions or deletions in that region that could indicate significant disruptions of the helical structure, as shown for *E. coli* and *F. tularensis* (Fig 9).

**Framework for a catalytic mechanism.** Combining the elements discussed above, we would like to suggest the hypothetical catalytic mechanism for the class II *Ft*FBPase illustrated in Fig 12. It highlights the importance of the OH nucleophile in the side chain of Thr89, activated by a water molecule in the coordination sphere of the high affinity Mn²⁺ cation. Probably, a short-lived covalent Thr-P intermediate would follow which would be cleaved by a water

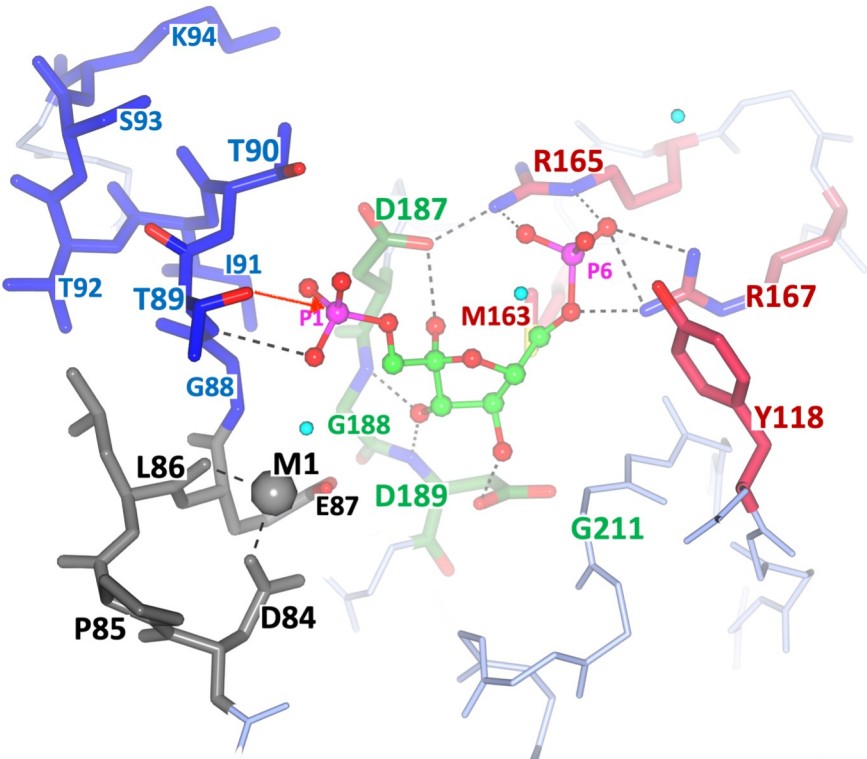

**Fig 9. Structural components of the catalytic mechanism of *Ft*(Mn²⁺)FPBaseII.** 3D rendering of the components of the catalytic mechanism for one-metal (Mn$^{2+}$) phosphate hydrolysis in *Ft*FBPaseII. Gray: Contiguous M1 site 84DPLE87. Green: Substrate binding, comprising the P6 pocket (conserved 165RDR167, cyan) and the conserved 187DGD189 sequence interacting with the three furanose ring hydroxyls. Thr89 OH nucleophile in front of the P1 cleavable phosphate is the beginning of the N-terminus of the helix segment providing the positive charge to stabilize the leaving group. M1 marks the position of the high affinity Mn$^{2+}$ divalent cation. Red arrow indicates the suggested attacking nucleophile to the cleavable P1. The residue E57 providing the lower apical octahedral ligand of M1 is not shown to provide a clear view of the furanose hydroxyl groups (S1 Fig).

molecule from the solvent, replacing the water that initiated the reaction. Characterization of the Thr-P reaction intermediate would be difficult. Several crystallographic experiments (forms D, and E) were attempted to capture its presence in the active site but did not succeed given the time scale of the crystallographic work, even with the slower T89S mutant. Mass spectroscopy experiments were considered impractical given the size of the tetrameric aggregate.

The detailed number and position of the divalent cations in the active Mg$^{2+}$-dependent FBPasesII such as *Mt*FBPaseII, as well as their coordination spheres, is still uncertain given the limited value of the available PDB structures (6ayy, 6ayu, 6ayv, and homologous entries) as indicated in the introduction. However, the structural insights derived from this work strongly suggest that activated water(s) molecules via the divalent cation(s) (Mn$^{2+}$or Mg$^{2+}$) combined with the hydroxyl group of Thr89 (or homologous e. g. Ser89) residue would be required, for catalysis in all class II FBPases. In addition, it is reasonable to assume that the conserved N-terminal helical turn dipole would also play a role in the stabilization of the transition phosphate intermediate in the Mg$^{2+}$-dependent subgroup of FBPasesII, based on the conservation of the helical feature (Fig 8). Further structural and biochemical studies on other members of the class II FBPase enzymes are planned to confirm this mechanism.

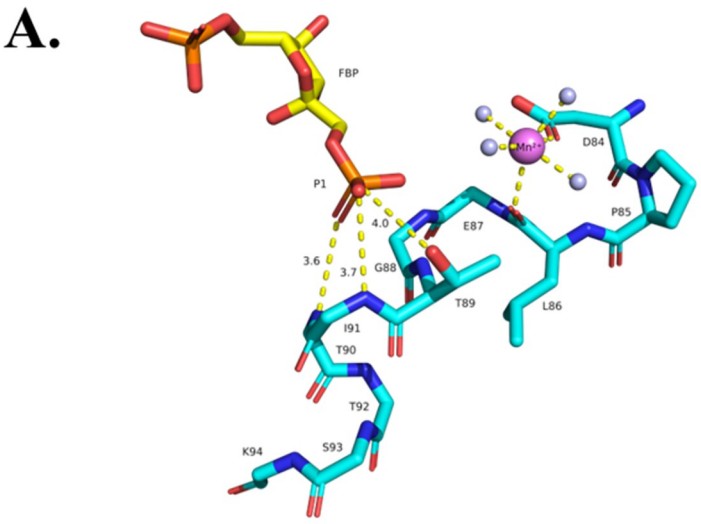

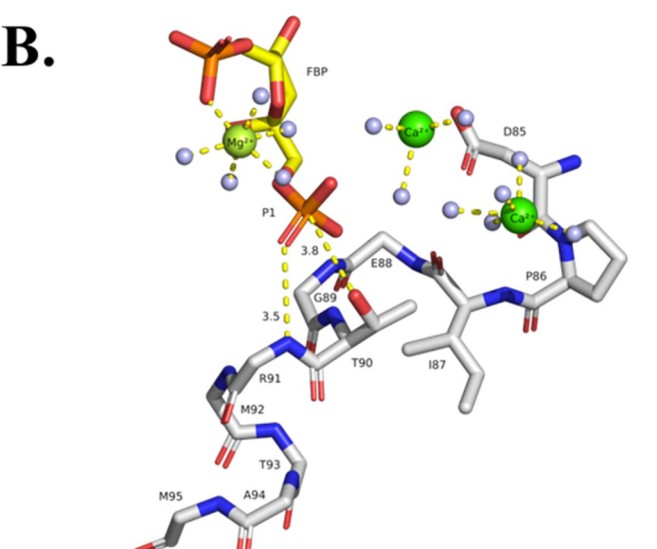

**Fig 10. Proposed stabilization of the leaving phosphate group (P1) during the catalytic process for class II FBPases.** (A) Ligands coordination by active *Ft*FBPaseII and (B) inactive *Ec*FBPaseII (PDB 3d1r). The interatomic distances and the side-chain important amino acid residues have been highlighted. Waters are shown as light blue spheres. Note the hydrogen bond interactions of the leaving phosphate group (P1) with the amide protons of the N-terminal helix in both structures. The presence of L86 (I87 in *E. coli*, **B**) probably orients and positions optimally the side chain OH group of T89 for the catalytic action. In *Ft*FBPase the T89S mutation would result in a less restrained OH yielding an enzyme with a lower catalytic activity; the T89A mutation resulted in a totally inactive enzyme [7]. Image in B) extracted from PDB entry 3d1r (*E. coli* FBPaseII) showing the homologous active site residues (see below). The structure contains two $Ca^{2+}$ and one $Mg^{2+}$ divalent cations in the proximity of the active and yet the enzyme is active almost exclusively with $Mn^{2+}$. The exact position of the divalent cation in the active enzyme is unknown. The only $Mg^{2+}$ is coordinated by the oxygens of the two phosphoryl groups in F1,6BP. In the *E. coli* FBPaseII structure (3d1r), the two $Ca^{2+}$ metals are 5.6 Å apart and are relatively far (4.7 and 6.6 Å) from the nearest phosphoryl oxygens. The images show with dashed lines the suggested nucleophilic attack of the Thr residue to the cleavable Phosphate (P1) in both enzymes (distances 3.8–4.0 Å).

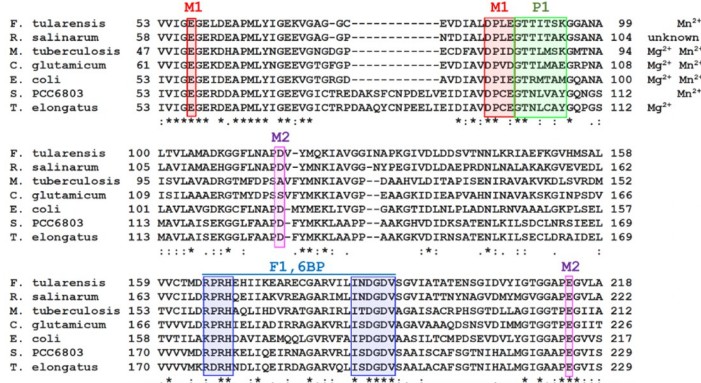

**Fig 11. Sequence alignment of several class II FBPases.** The selected regions include: Red and pink—the highly conserved regions around the critical high affinity metal sites (M1 and M2). Green, Thr89 (*Ft*FBPaseII) residues and the following two-turn helix whose helical dipole is suggested to be implicated in the catalytic mechanism. Blue—the extended conserved sequence beginning with the Arg-Pro-Arg motif that interacts with the anchored P6, and the conserved IxDGD motive approximately twenty residues downstream that is critical for interactions with the hydroxyl groups of the furanose ring of the F1,6BP substrate. Divalent cations required for activity are listed, although the catalytic activities vary significantly at different concentrations as reported in the literature and discussed above. The selected regions encompass the contiguous highly conserved N-terminal domains (two thirds (218/328 amino acids) of the structures of a wide variety of FBPasesII enzymes with a percent sequence identity ranging from 40–48%. The sequence identity of Class I (e.g. 2q8m) and Class IV (e.g 1dka) FBPases against ClassII is about 20–25%). The F1,6BP binding finger print mentioned above is not present in the amino acid sequences of the other classes.

## Conclusions

We have been able to infer a consistent binding mode of the F1,6BP substrate in the active site pocket of class II FBPases in the presence of the native metal cofactors and surrounding water molecules. This was accomplished by analysis of the available structural information from previous publications and PDB deposited structures of class II FBPases from *E. coli*, *M. tuberculosis*, and *F. tularen*sis, combined with the five additional crystal structures presented here. These include active *Ft*FBPaseII complexed with a native metal cofactor $Mn^{2+}$ and a partially occupied F6P product, together with the structure of substrate bound in *Mt*FBPaseII and three new *Ft*FBPaseII crystal structures reported here. This allowed us to propose a framework for the catalytic mechanism for the entire class II FBPases, consistent with conserved structural features in the proximity of the active site for the enzymes of the three species used here for analysis. Namely, the presence of a critical hydroxyl-containing residue (Thr/Ser) that provides the critical nucleophile that attacks the cleavable phosphate-1 of the substrate and the existence of an N-terminal helical turn that provides a dipole-induced positive charge to stabilize the transition state of the cleavable phosphate-1. Although the suggested catalytic mechanism has not been proven in this work, the proposed framework is more consistent chemically and structurally than the earlier proposals based on the initial *E. coli* structure (PDB 3d1r). We hypothesize that this mechanism will also be applicable to other class members based on the high conservation of the amino acid sequences around those structural features. The specific native divalent cation may vary, most commonly $Mg^{2+}$ or $Mn^{2+}$. However, the critical water coordination sphere around the native divalent cation, and the role of the conserved helical dipole would remain the key structural features of the enzymatic mechanism. Our results are consistent with the requirement of only one metal site for activity in class II FBPases. However, a second cation binding site may exist at higher concentrations, but it may hinder rather than enhance the enzymatic activity, or could induce a different two-metal dependent catalytic

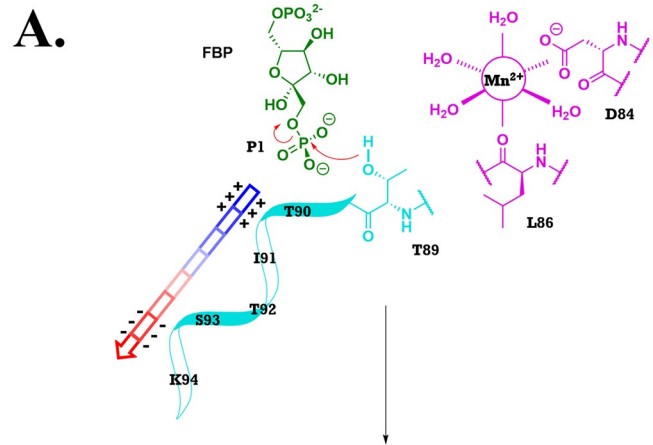

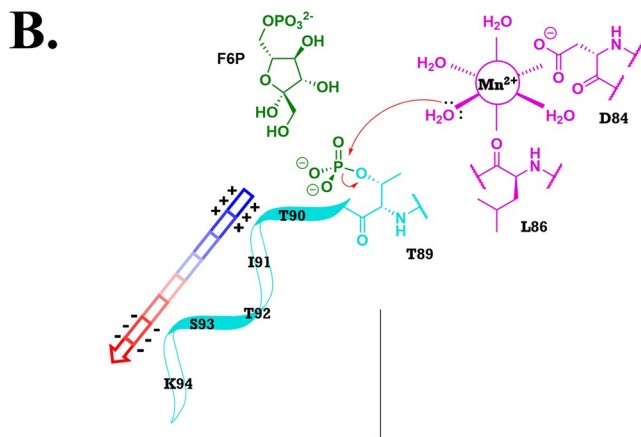

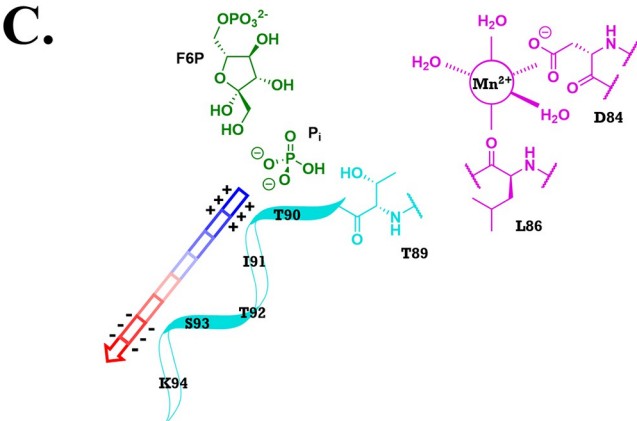

**Fig 12. Details of the hypothesized catalytic mechanism based on the framework proposed.** (A) Cleavage of phosphate-1 and stabilization of the transition state intermediate by the adjacent two turn N-terminus α-helix. (B) Hydrolysis of the Thr89-OPO$_3^{2-}$ intermediate by a Mn$^{2+}$ coordinated water molecule to release an inorganic phosphate. (C) Overall product of reaction. Proton transfers are implied.

mechanism, whereby the enzymes would catalyze the reaction at different reaction rates. The biological relevance (if any) of this possibility is unknown.

## Supporting information

**S1 Fig. Mn$^{2+}$ coordination in the active site of Chain C in the structure of *Ft(*Mn$^{2+}$) FBPaseII-Glycerol/F6P, Form A (Chain C).** 2F$_o$-F$_c$ electron density map contoured at 1.5σ. (TIF)

**S2 Fig. Omit (F$_o$-F$_c$) electron density map of the refined structure of *Mt*FBPaseII (Form C).**
(TIF)

**S3 Fig. Active site of Form D *Ft*FBPase crystals (8g5w).**
(TIF)

**S1 Table. Ligands in *Ft*(Mn$^{2+}$)FBPase-F6P (7txg).**
(DOCX)

## Acknowledgments

The Advanced Photon Source was supported by the US Department of Energy (DE-AC02-06CH11357). In addition, we acknowledge the help of the LS-CAT staff at the Advanced Photon Source, Argonne National Laboratory, with data collection on the LS-CAT 21-ID beamlines. The assistance of Dr. Kiira Ratia in the preparation of the figures is appreciated. The authors would also like to thank Professor Scott Franzblau for supporting this project under the auspices of the Institute of Tuberculosis Research.

## Author Contributions

**Conceptualization:** Anna I. Selezneva, Farahnaz Movahedzadeh, Celerino Abad-Zapatero.

**Data curation:** Anna I. Selezneva, Luke N. M. Harding, Hiten J. Gutka, Celerino Abad-Zapatero.

**Formal analysis:** Anna I. Selezneva, Celerino Abad-Zapatero.

**Funding acquisition:** Farahnaz Movahedzadeh, Celerino Abad-Zapatero.

**Investigation:** Celerino Abad-Zapatero.

**Methodology:** Anna I. Selezneva.

**Resources:** Hiten J. Gutka, Farahnaz Movahedzadeh, Celerino Abad-Zapatero.

**Supervision:** Farahnaz Movahedzadeh.

**Writing – original draft:** Anna I. Selezneva, Luke N. M. Harding, Farahnaz Movahedzadeh, Celerino Abad-Zapatero.

**Writing – review & editing:** Anna I. Selezneva, Luke N. M. Harding, Hiten J. Gutka, Farahnaz Movahedzadeh, Celerino Abad-Zapatero.

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
