## [Decision Letter · Decision Letter 0]

28 Oct 2022

PONE-D-22-24612New Structure of Class II Fructose-1,6-Bisphosphatase from Francisella tularensis suggest a novel catalytic mechanism for the entire classPLOS ONE

Dear Dr. Abad-Zapatero,

Thank you for submitting your manuscript to PLOS ONE. After careful consideration, we feel that it has merit but does not fully meet PLOS ONE’s publication criteria as it currently stands. Therefore, we invite you to submit a revised version of the manuscript that addresses the points raised during the review process.

I apologize for the delay with reviewing this manuscript. As you will see, one of the reviewers in particular suggests that some interpretations are not supported by the data, and also that the structures can be improved further. Please consider the comments and provide a point-by-point response and description of revisions you made when you submit the revised manuscript.

We look forward to receiving your revised manuscript.

Kind regards,

Bostjan Kobe, Ph.D.

Academic Editor

PLOS ONE

Journal Requirements:

Reviewers' comments:

Reviewer's Responses to Questions

**Comments to the Author**

1. Is the manuscript technically sound, and do the data support the conclusions?

Reviewer #1: Partly

Reviewer #2: Partly

2. Has the statistical analysis been performed appropriately and rigorously? 

Reviewer #1: Yes

Reviewer #2: N/A

3. Have the authors made all data underlying the findings in their manuscript fully available?

Reviewer #1: Yes

Reviewer #2: Yes

4. Is the manuscript presented in an intelligible fashion and written in standard English?

Reviewer #1: Yes

Reviewer #2: Yes

5. Review Comments to the Author

Reviewer #1: The paper entitled “New Structure of Class II Fructose-1,6-Bisphosphatase from Francisella tularensis

suggest a novel catalytic mechanism for the entire class” presents structures of FBPase II

from Francisella tularensis resolved using the crystallographic technique. According to the authors, the structures represent the active state of the enzyme, containing metal cofactor Mn2+ and complexed with the product of the reaction fructose-6-phosphate (F6P), and the substrate fructose-1,6-bisphosphate (F1,6P2). Their analysis of the structures presented in the manuscript suggests a novel mechanism of catalysis for FBPases II.

The manuscript is clearly written and demonstrates a convincing set of data supporting the conclusions. The object of study (FBPase II from Francisella tularensis) is not only interesting for biochemists/biologists but it may be also important from medical and public safety points of view.

While I have no doubts that the manuscript is acceptable for publication, I have some questions concerning conditions of the crystallization. The authors speculate about the mechanism of the reaction based on structures achieved by crystallization in the presence of very high concentrations of the product and the substrate of the reaction. However, it has been shown that high titer of the substrate can “inhibit” the rate of the reaction. It has been hypothesized that F1,6P2 at high concentrations binds to mammalian FBPases in slightly different manner than it does at low concentrations and this results in a lower kcat (“b x kcat, where b<1). I do not know if such “inhibition by substrate” has ever been observed in the case of FBPase II, but if it has, the structures resolved by the authors represent the enzyme which catalytic site is occupied by the substrate in a configuration suboptimal for catalysis.

Obviously, it is still very valuable structure but the mechanism of catalysis may be slightly different.

On the other hand, Francisella tularensis is an intracellular pathogen and the titer of F1,6P2 in human cells is about 30-50 microM, pH is close to 7.0 and the temperature is ~37C (~310K). Why have the authors measured the kinetic parameters using 293K, pH 8.0 and 400 microM F1,6P2? I could not find any information about km and/or Kd for F1,6P2 and FBPase II from Francisella tularensis throughout the paper.

The kinetic studies were performed at pH 8.0 while the crystallization was carried out at pH 6.0. Does pH have no effect on the arrangement of the FBPase II active site?

I believe that a short discussion of the above problems could clarify the concerns about the results.

Reviewer #2: RE: PONE-D-22-24612

Dear Editor,

The manuscript: New Structure of Class II Fructose-1,6-Bisphosphatase from Francisella tularensis suggest a novel catalytic mechanism for the entire class by Abad-Zapatero et al. presents a three structures of FBPaseII from two different species with ligands. This work is aimed at elucidating the details of the catalytic reaction mechanism that is still under controversy for this class of FBPases. This paper presenting three new complexes of previously published structures, and additional results and interpretations, certainly deserves to be published. Unfortunately, this referee found significant problems with the structures, significant problems with their interpretation as well as with the proposals for the mechanism drawn. In its present form the paper is not suitable for publication.

In the referee opinion several alternative/synergistic actions need to be taken.

1) The structures need to be improved.

The structure 7txa has deeply unfinished water structure that is coupled with unexplained densities in the majority of active sites. The two 6FP refined are doubtful which problem is entangled with identification of the metal ion binding sites. The best example is provided by inspection of temperature factors of 6FP in which they vary by more than two fold in a single molecule. Two fragments of the backbone, 57-65 and 322-328 are systematically misplaced. All have weaker densities in all eight subunits. They nevertheless can be properly traced by gradual improvement of phases (adding proper elements to the model) and by gradual lowering of the contouring level of ED. The placement of phosphates in F6Ps are doubtful.

In the structure 7txb backbone is relatively well placed but contains numerous conformational errors (peptide flips) that are the most visible by comparison between subunits. Additionally I believe that they are indicators of alternative locations for 109-113 region. The subunits are widely separated and do not form any biological unit. They should be brought together for reliable modelling by crystallographic symmetries to form a biologically relevant unit. Too many water molecules at 3.7Å combined with a flipped configuration between two refined FBP (6P in place of 1P in subunit A) make the findings unreliable.

In the structure 7txg which is of the highest resolution and constitutes the basis for the entire publication the uncertainty in identification of metal ions and lack of full description of their coordination spheres makes the speculations very problematic. The best defined active site in subunit C contains most likely a product F6P intermixed/overlapping with the glycerol. My initial attempt at resolving ambiguities lead to 70% F6P and 30% glycerol. So a combination of findings in corrected 7txg and 7txb combined with E.coli structure (3d1r, that also unfortunately contains some errors, the placement of the head group of Glu213) might constitute a useful starting point for speculations about the catalytic mechanism. Unfortunately, suggesting that off-axial remote water molecule as initiating a reaction is outside of the realm of possibility, when existing literature is taken into account, and needs to be either excised or alternatively supported with solid experimental evidence.

2) More thorough interpretation based on a solid literature review need to be presented in the introduction and thrown back on a broader chemical and biochemical background that needs to be cited. It means that the discussion of the possible pathways for phosphate hydrolysis must be reviewed and cited. As a useful citation authors can use: Fundamentals of Phosphate Transfer by Anthony J. Kirby and Faruk Nome Acc. Chem. Res. 2015, 48, 1806−1814 (or any other classic works including that of the editor). This citation explains that even though a nucleophile can attack from any direction cannot be a priori excluded, the preferred one is carried out in the “in-line” direction along the connecting bond. Some experimental classic works can be cited also (Richard Honzatko or Karen Allen). Recalling the reference 4 (Brown G, Singer A, Lunin VV, Proudfoot M, Skarina T, Flick R, et al. Structural and biochemical characterization of the type II fructose-1,6-bisphosphatase GlpX from Escherichia coli. J Biol Chem. 2009;284(6):3784-92) is obviously necessary but needs to be treated with caution. This publication misrepresents the existing literature as well as misinterprets their own structure. The same mistake in my opinion have been committed by the authors of this publication. Neither structures, whether E. coli, or MT, or FT, support the notion that the water molecule is an attacking nucleophile. A much more likely nucleophile in all three species is activated Thr89/90 Oγ. Such a hypothesis should be thoroughly evaluated though, as it invokes a covalent intermediate and can be relatively easily probed. All schemes in the paper as well as most of the figures are deeply misleading as they are directed to support a very low probability pathway.

3) Alternatively the paper can be formulated as a structural report without speculating about the nature of the catalytic mechanism. However, if the authors select discussing this issues it needs to be tested. There are several tests that can be applied here. Firstly, one can test for the presence of phosphorylated Thr by use of appropriate antibodies. Secondly, one can check by isotope exchange the presence of appropriate labelled atoms (deuterium, or labeled oxygen) by many different methods (NMR, Mas Spec, fast X-ray etc). Another method is isotope exchange kinetic method. Unfortunately, the role of a referee is not to direct the necessary science, so I am only sharing some advice to guide how to achieve the publishable manuscript that would meet the scrutiny of this referee. The authors need to commit themselves to a particular solution and improve the manuscript in the spirit of the chosen direction.

In essence the paper cannot be published in its present form but may become publishable after major corrections.

Detailed remarks.

1) The paper is relatively well written and edited. However, there are many places that can be improved. For instance, I am confused with numbering and placement of individual Figures. I would recommend including Table S1 with refinement statistics inside the main body of the paper. Nowhere in the text, besides TableS1 the estimation of quality nor final statistics are mentioned.

2) The abstract must be adapted to the version of the paper authors select to publish. I absolutely agree with the proposal that the helical motif 88-94 is stabilizing the leaving phosphate and contributes to its binding but the statements about a nucleophile must be clarified or removed. Additionally arguments about activity of mutants or combination of metals must be tempered. It is very well documented that mobile loops play significant role in activity of all FBPases. Secondly, it is very well documented that different metal ions play different roles and that the inhibitory metal ion binding might be an activating factor. Multiple examples can be found in the literature. This point must be dealt with appropriate subtlety. I share the criticism of previous publications but my criticism persist towards what was written in the introduction.

3) The Figures must be reorganized to form a more logical story, it means renumber and relocate them. The Figure captions must be improved to provide more specific content. For instance, the subunit and the contouring level of the type of the ED used in the figure, must be mentioned

4) Kinetics presented in Fig2 are very interesting but unequivocally indicate that at least two metal ions are involved in catalysis as indicated by the burst phase, no matter whether only one or two types of metal ions are involved. A useful guide is also inhibitory phase that usually happens when a leaving group is associated with the separate associated metal ion. More physicochemical studies would be needed to establish the number and affinities for individual metal ion binding sites, for instance using DSC or by plasmon resonance.

5) Identification of many glycerol molecules is doubtful. Some kind of uniform test should be applied to produce more reliable identification of solute components. The claim that glycerol molecules interfere with refinement of F6P is an unjustified/unfounded technical obstacle. In Coot/Refmac it is enough to use disordered A,B,C components to obtain unreacting/nonrepelling/overlapping models representing partial occupancies.

6) Fig6B is extremely misleading. Not only placement of F6P is doubtful in form B but also the view is such that diffuses the impression that 6P is closer to the position of 1P in the substrate as defined in form B.

7) I like Fig9 in design but it needs to be improved providing that the correct ions with correct coordination spheres are depicted. See my previous doubts. Did authors used anomalous signal to detect their identities as the variable occupancy might be misleading during the refinement?

8) Fig10 presents an impossibility, at least as the chemical consensus stands currently, about the phosphotransfer reactions, and needs to be changed or deleted.

9) The Supporting information has to be edited appropriately and at present time has several small defects. For instance in Table SI2 the info about the temperature factors is muddled. Most likely the first and the last column represent minima and maxima of Bs, not as described the third and fourth value. Also RSCC cited do not provide full confidence about the proper identification of ligands. I am also abstracting from a fact that the numerical descriptions are different from the coordinate files I obtained. For instance temperature factors of Mn ions in my file of form A are 53, 145, 118, 129 which certainly are different from 53, 93, 73, 73 cited in the table. Please be consistent. I do not even mention that any attempt at refinement changes these numbers significantly.

6. PLOS authors have the option to publish the peer review history of their article (what does this mean?). If published, this will include your full peer review and any attached files.

Reviewer #1: No

Reviewer #2: No

---

## [Author Response · Author response to Decision Letter 0]

31 Mar 2023

5. Review Comments to the Author

There is a significant change in the Title of the revised manuscript that better reflects the new content and the new emphasis.

Revised titled:

New structures of Class II Fructose-1,6-Bisphosphatase from Francisella tularensis provide a framework for a novel catalytic mechanism for the entire class 

Reviewer #1: The paper entitled “New Structure of Class II Fructose-1,6-Bisphosphatase from Francisella tularensis

suggest a novel catalytic mechanism for the entire class” presents structures of FBPase II

from Francisella tularensis resolved using the crystallographic technique. According to the authors, the structures represent the active state of the enzyme, containing metal cofactor Mn2+ and complexed with the product of the reaction fructose-6-phosphate (F6P), and the substrate fructose-1,6-bisphosphate (F1,6P2). Their analysis of the structures presented in the manuscript suggests a novel mechanism of catalysis for FBPases II.

The manuscript is clearly written and demonstrates a convincing set of data supporting the conclusions. The object of study (FBPase II from Francisella tularensis) is not only interesting for biochemists/biologists but it may be also important from medical and public safety points of view.

While I have no doubts that the manuscript is acceptable for publication, I have some questions concerning conditions of the crystallization. The authors speculate about the mechanism of the reaction based on structures achieved by crystallization in the presence of very high concentrations of the product and the substrate of the reaction. However, it has been shown that high titer of the substrate can “inhibit” the rate of the reaction. It has been hypothesized that F1,6P2 at high concentrations binds to mammalian FBPases in slightly different manner than it does at low concentrations and this results in a lower kcat (“b x kcat, where b<1). I do not know if such “inhibition by substrate” has ever been observed in the case of FBPase II, but if it has, the structures resolved by the authors represent the enzyme which catalytic site is occupied by the substrate in a configuration suboptimal for catalysis.

>We are aware of those issues regarding the ‘inhibition by substrate’ in Class I (mammalian) phosphatases. We have not observed such an effect in our ClassII FBPases examples (F. tularensis, M. tuberculosis). We have emphasized the differences between ClassI and ClassII phosphatases in the revised introduction. We disagree with the second comment regarding the ‘suboptimal configuration for catalysis’. Previously, we have never come to that conclusion from now previous work. In addition, our new structure of the FtFBPaseII bound to the substrate (under suboptimal Mn2+ concentration, new structure Form E, 8g5x), show that the binding of the substrate in ClassII FBPases is unique and distinct to the class II, resulting in a different catalytic mechanism, as outlined in the revised version of the manuscript, as submitted. 

Obviously, it is still very valuable structure but the mechanism of catalysis may be slightly different.

On the other hand, Francisella tularensis is an intracellular pathogen and the titer of F1,6P2 in human cells is about 30-50 microM, pH is close to 7.0 and the temperature is ~37C (~310K). Why have the authors measured the kinetic parameters using 293K, pH 8.0 and 400 microM F1,6P2? I could not find any information about km and/or Kd for F1,6P2 and FBPase II from Francisella tularensis throughout the paper.

The kinetic studies were performed at pH 8.0 while the crystallization was carried out at pH 6.0. Does pH have no effect on the arrangement of the FBPase II active site?

I believe that a short discussion of the above problems could clarify the concerns about the results.

>Comments about these issues have been included in the revised version. 

Reviewer #2: RE: PONE-D-22-24612

Dear Editor,

The manuscript: New Structure of Class II Fructose-1,6-Bisphosphatase from Francisella tularensis suggest a novel catalytic mechanism for the entire class by Abad-Zapatero et al. presents a three structures of FBPaseII from two different species with ligands. This work is aimed at elucidating the details of the catalytic reaction mechanism that is still under controversy for this class of FBPases. This paper presenting three new complexes of previously published structures, and additional results and interpretations, certainly deserves to be published. Unfortunately, this referee found significant problems with the structures, significant problems with their interpretation as well as with the proposals for the mechanism drawn. In its present form the paper is not suitable for publication.

In the referee opinion several alternative/synergistic actions need to be taken.

1) The structures need to be improved.

The structure 7txa has deeply unfinished water structure that is coupled with unexplained densities in the majority of active sites. The two 6FP refined are doubtful which problem is entangled with identification of the metal ion binding sites. The best example is provided by inspection of temperature factors of 6FP in which they vary by more than two fold in a single molecule. Two fragments of the backbone, 57-65 and 322-328 are systematically misplaced. All have weaker densities in all eight subunits. They nevertheless can be properly traced by gradual improvement of phases (adding proper elements to the model) and by gradual lowering of the contouring level of ED. The placement of phosphates in F6Ps are doubtful.

In the structure 7txb backbone is relatively well placed but contains numerous conformational errors (peptide flips) that are the most visible by comparison between subunits. Additionally I believe that they are indicators of alternative locations for 109-113 region. The subunits are widely separated and do not form any biological unit. They should be brought together for reliable modelling by crystallographic symmetries to form a biologically relevant unit. Too many water molecules at 3.7Å combined with a flipped configuration between two refined FBP (6P in place of 1P in subunit A) make the findings unreliable.

In the structure 7txg which is of the highest resolution and constitutes the basis for the entire publication the uncertainty in identification of metal ions and lack of full description of their coordination spheres makes the speculations very problematic. The best-defined active site in subunit C contains most likely a product F6P intermixed/overlapping with the glycerol. My initial attempt at resolving ambiguities lead to 70% F6P and 30% glycerol. So a combination of findings in corrected 7txg and 7txb combined with E.coli structure (3d1r, that also unfortunately contains some errors, the placement of the head group of Glu213) might constitute a useful starting point for speculations about the catalytic mechanism. Unfortunately, suggesting that off-axial remote water molecule as initiating a reaction is outside of the realm of possibility, when existing literature is taken into account, and needs to be either excised or alternatively supported with solid experimental evidence.

>Structures for PDB entries have been revised and improved and new structures are available from the PDB (validation reports are included: 7txa, 7txb, 7txg). In addition, two new structures have been added and deposited in the PDB of excellent quality in terms of data completeness and refinement statistics (8g5x,8g5w). In addition, it has been emphasized in the current manuscript that the low symmetry of the crystal structures of FtFBPaseII (P1, in all four crystal forms), with a full tetramer as a biological and crystallographic assembly results in 20 independent active site environments. Significant variations were found due crystal packing effects, different crystallization conditions, divalent cations concentrations etc., as discussed in the manuscript. The detailed analysis of these various active sites made us attempt to infer a chemical and catalytically consistent structure for the ‘active’ enzyme-substrate complex and outline the elements of catalysis as suggested. There are individual variations in each of the different ‘active site’ that cannot be discussed and are not relevant to this manuscript. 

2) More thorough interpretation based on a solid literature review need to be presented in the introduction and thrown back on a broader chemical and biochemical background that needs to be cited. It means that the discussion of the possible pathways for phosphate hydrolysis must be reviewed and cited. As a useful citation authors can use: Fundamentals of Phosphate Transfer by Anthony J. Kirby and Faruk Nome Acc. Chem. Res. 2015, 48, 1806−1814 (or any other classic works including that of the editor). This citation explains that even though a nucleophile can attack from any direction cannot be a priori excluded, the preferred one is carried out in the “in-line” direction along the connecting bond. Some experimental classic works can be cited also (Richard Honzatko or Karen Allen). Recalling the reference 4 (Brown G, Singer A, Lunin VV, Proudfoot M, Skarina T, Flick R, et al. Structural and biochemical characterization of the type II fructose-1,6-bisphosphatase GlpX from Escherichia coli. J Biol Chem. 2009;284(6):3784-92) is obviously necessary but needs to be treated with caution. This publication misrepresents the existing literature as well as misinterprets their own structure. The same mistake in my opinion have been committed by the authors of this publication. Neither structures, whether E. coli, or MT, or FT, support the notion that the water molecule is an attacking nucleophile. A much more likely nucleophile in all three species is activated Thr89/90 Oγ. Such a hypothesis should be thoroughly evaluated though, as it invokes a covalent intermediate and can be relatively easily probed. All schemes in the paper as well as most of the figures are deeply misleading as they are directed to support a very low probability pathway.

>We have revised the section on the catalytic mechanism addressing the elements of the catalysis of phosphoryl transfer as outlined in the literature. Based on the new structures (PDB entries 8g5w, 8g5x) we have shown that the constellation of waters in the absence of substrate is essentially displaced by substrate binding, resulting in a slightly different mode of binding of the cleavable phosphate (Phosphate 1), that -without any ‘modelling’ – binds right in front of the now proposed Thr89-OH nucleophile. The various figures have been revised and are now consistent with this novel catalytic hypothesis. Attempts to characterize the short lived ‘covalent intermediate’ by crystallographic or MS protocols during the time allowed for this revision have failed. Future attempts will use our slower T89S mutant on the Ft or Mt enzymes. 

3) Alternatively the paper can be formulated as a structural report without speculating about the nature of the catalytic mechanism. However, if the authors select discussing this issues it needs to be tested. There are several tests that can be applied here. Firstly, one can test for the presence of phosphorylated Thr by use of appropriate antibodies. Secondly, one can check by isotope exchange the presence of appropriate labelled atoms (deuterium, or labeled oxygen) by many different methods (NMR, Mas Spec, fast X-ray etc). Another method is isotope exchange kinetic method. Unfortunately, the role of a referee is not to direct the necessary science, so I am only sharing some advice to guide how to achieve the publishable manuscript that would meet the scrutiny of this referee. The authors need to commit themselves to a particular solution and improve the manuscript in the spirit of the chosen direction.

In essence the paper cannot be published in its present form but may become publishable after major corrections.

>Based on these extensive revisions and the important addition of two new structures supporting the most probable mechanism, we would still like the manuscript to be considered for a full research manuscript.

Detailed remarks.

1) The paper is relatively well written and edited. However, there are many places that can be improved. For instance, I am confused with numbering and placement of individual Figures. I would recommend including Table S1 with refinement statistics inside the main body of the paper. Nowhere in the text, besides TableS1 the estimation of quality nor final statistics are mentioned.

>The Table with data collection and refinement statistics for the 5 structures discussed (3 previous version plus 2 new ones) is now part of the main body of the manuscript. It has been removed from the Supplementary information. 

2) The abstract must be adapted to the version of the paper authors select to publish. I absolutely agree with the proposal that the helical motif 88-94 is stabilizing the leaving phosphate and contributes to its binding but the statements about a nucleophile must be clarified or removed. Additionally arguments about activity of mutants or combination of metals must be tempered. It is very well documented that mobile loops play significant role in activity of all FBPases. Secondly, it is very well documented that different metal ions play different roles and that the inhibitory metal ion binding might be an activating factor. Multiple examples can be found in the literature. This point must be dealt with appropriate subtlety. I share the criticism of previous publications but my criticism persist towards what was written in the introduction.

>We have modified the title and content of the revised manuscript to identify the basic elements for catalysis and present the mechanism only as a hypothesis consistent with the available published literature and our extensive new data. We have revised the introduction to qualify the differences among FBPases. Also, our work as described in the manuscript documents that the way the substrate binds in the ClassII phosphatases is unique. Our work documents variations on the 55-65 amino acid loop (ordered/disordered) regarding packing and absence/presence in various active sites environments (of the 20 available). The best-defined structure of this loop in the ‘canonical’ divalent cation binding site (M1) has been documented. 

3) The Figures must be reorganized to form a more logical story, it means renumber and relocate them. The Figure captions must be improved to provide more specific content. For instance, the subunit and the contouring level of the type of the ED used in the figure, must be mentioned

>Figures have been rearranged, renumbered, improved and simplified for easier understanding. Figure captions have also been improved. 

4) Kinetics presented in Fig2 are very interesting but unequivocally indicate that at least two metal ions are involved in catalysis as indicated by the burst phase, no matter whether only one or two types of metal ions are involved. A useful guide is also inhibitory phase that usually happens when a leaving group is associated with the separate associated metal ion. More physicochemical studies would be needed to establish the number and affinities for individual metal ion binding sites, for instance using DSC or by plasmon resonance.

>The new structures document that at high concentrations of the divalent cation Mn2+ (and Mg2+, Selevneva et al. Acta Cryst. 2020), FtFBPase and possibly other members of the class can bind at two sites in the active site (see Fig.6B). Our activity data and other reports are consistent with the suggestion that the two-metal bound enzyme could have a different reaction rate. This needs to be investigated further. 

5) Identification of many glycerol molecules is doubtful. Some kind of uniform test should be applied to produce more reliable identification of solute components. The claim that glycerol molecules interfere with refinement of F6P is an unjustified/unfounded technical obstacle. In Coot/Refmac it is enough to use disordered A,B,C components to obtain unreacting/nonrepelling/overlapping models representing partial occupancies.

>The detailed discussion of glycerol sites has been removed from the main body of the manuscript. Only the relevant glycerol at the active site to show the partial occupancy has been retained. The various glycerol molecules in the structures have been revised in the updated PDB entries. 

6) Fig6B is extremely misleading. Not only placement of F6P is doubtful in form B but also the view is such that diffuses the impression that 6P is closer to the position of 1P in the substrate as defined in form B.

7) I like Fig9 in design but it needs to be improved providing that the correct ions with correct coordination spheres are depicted. See my previous doubts. Did authors used anomalous signal to detect their identities as the variable occupancy might be misleading during the refinement?

>The crystallography work did not look for anomalous signals as there was only one divalent cation (Mn2+) in all the crystallographic work on FtFBPase included in this manuscript. Only Mn2+ was present in all the crystallographic experiments.

8) Fig10 presents an impossibility, at least as the chemical consensus stands currently, about the phosphotransfer reactions, and needs to be changed or deleted.

>The Figure(s) outlining the hypothesis for the catalytic mechanism have been revised and updated.

9) The Supporting information has to be edited appropriately and at present time has several small defects. For instance in Table SI2 the info about the temperature factors is muddled. Most likely the first and the last column represent minima and maxima of Bs, not as described the third and fourth value. Also RSCC cited do not provide full confidence about the proper identification of ligands. I am also abstracting from a fact that the numerical descriptions are different from the coordinate files I obtained. For instance temperature factors of Mn ions in my file of form A are 53, 145, 118, 129 which certainly are different from 53, 93, 73, 73 cited in the table. Please be consistent. I do not even mention that any attempt at refinement changes these numbers significantly.

>The supplementary information now contains only one Table (Table SI1) that has been updated with the information for the revised PDB entry 7txg. The data collection and refinement statistics are now part of the main body of the manuscript.

---

## [Decision Letter · Decision Letter 1]

26 May 2023

PONE-D-22-24612R1New structures of Class II Fructose-1,6-Bisphosphatase from Francisella tularensis provide a framework for a novel catalytic mechanism for the entire classPLOS ONE

Dear Dr. Abad-Zapatero,

Thank you for submitting your manuscript to PLOS ONE. After careful consideration, we feel that it has merit but does not fully meet PLOS ONE’s publication criteria as it currently stands. Therefore, we invite you to submit a revised version of the manuscript that addresses the points raised during the review process.

 I asked one of the reviewers to check if they are happy with the revised version, and they suggested fixing some minor inconsistencies before it is accepted.

We look forward to receiving your revised manuscript.

Kind regards,

Bostjan Kobe, Ph.D.

Academic Editor

PLOS ONE

Journal Requirements:

Reviewers' comments:

Reviewer's Responses to Questions

**Comments to the Author**

1. If the authors have adequately addressed your comments raised in a previous round of review and you feel that this manuscript is now acceptable for publication, you may indicate that here to bypass the “Comments to the Author” section, enter your conflict of interest statement in the “Confidential to Editor” section, and submit your "Accept" recommendation.

Reviewer #2: (No Response)

2. Is the manuscript technically sound, and do the data support the conclusions?

Reviewer #2: (No Response)

3. Has the statistical analysis been performed appropriately and rigorously? 

Reviewer #2: Yes

4. Have the authors made all data underlying the findings in their manuscript fully available?

Reviewer #2: Yes

5. Is the manuscript presented in an intelligible fashion and written in standard English?

Reviewer #2: Yes

6. Review Comments to the Author

Reviewer #2: This version of the manuscript is significantly improved in form and in spirit. It avoids major discrepancies with published knowledge. It is partially suitable for publication. Several minor problems persist that require attention and correction. It however dos not require my supervision and further review.

The most significant problem is that the paper requires careful reading and correcting a number of typos. There are multiple instances of mistyped/misplaced letters, symbols and the most particularly unintended concatenations (lack of white spaces). There are unjustified use of long hyphens and similar editing problems. I found a very puzzling use of (Ref: Nishimasu-Structure-2004). Is this a real citation? It is not on references list. Additionally I found very puzzling the statement: "This results in a C�-C� distance between the chains of 1.5 A at this position." If it is true then it is a clear error. If it is a suspected artifact then it should not be mentioned as an observation. The safest method of dealing with this issue at this stage is to remove it. It does not add anything to the body of the paper.

7. PLOS authors have the option to publish the peer review history of their article (what does this mean?). If published, this will include your full peer review and any attached files.

Reviewer #2: No

---

## [Author Response · Author response to Decision Letter 1]

6 Jun 2023

My colleagues and I are pleased to submit the final revised version of the manuscript entitled ‘New structures of Class II Fructose-1,6-Bisphosphatase from Francisella tularensis provide a framework for a novel catalytic mechanism for the entire class’ for your consideration for publication in PLoS One as an original research article. Corresponding authors are Farahnaz Movahedzadeh and Celerino Abad-Zapatero. 

The revised version addresses the issues raised by one of the referees (#2) as follows. A reference was missing (Nishimasu, 2004-Structure) introducing the characteristics of Class V FBPases. This reference has been added and the corresponding sequence of references revised all through the text and in the final list. The controversial sentence containing ‘Ca-Ca’ distance has been removed. The complete text of this revised version has been carefully reviewed for typos and other minor editorial discrepancies.

---

## [Editor Report · Decision Letter 2]

8 Jun 2023

New structures of Class II Fructose-1,6-Bisphosphatase from Francisella tularensis provide a framework for a novel catalytic mechanism for the entire class

PONE-D-22-24612R2

Dear Dr. Abad-Zapatero,

We’re pleased to inform you that your manuscript has been judged scientifically suitable for publication and will be formally accepted for publication once it meets all outstanding technical requirements.

Kind regards,

Bostjan Kobe, Ph.D.

Academic Editor

PLOS ONE
---

## [Editor Report · Acceptance letter]

15 Jun 2023

PONE-D-22-24612R2 

New structures of Class II Fructose-1,6-Bisphosphatase from *Francisella tularensis* provide a framework for a novel catalytic mechanism for the entire class 

Dear Dr. Abad-Zapatero:

I'm pleased to inform you that your manuscript has been deemed suitable for publication in PLOS ONE. Congratulations! Your manuscript is now with our production department. 

Kind regards, 

on behalf of

Professor Bostjan Kobe 

Academic Editor

PLOS ONE